# Dynamic estimation of auditory temporal response functions via state-space models with Gaussian mixture process noise

Sina Miran[1], Alessandro Presacco[2], Jonathan Z. Simon[2,3,4], Michael C. Fu[2,5], Steven I. Marcus[2,3], Behtash Babadi[2,3]*

**1** Starkey Hearing Technologies, Eden Prairie, Minnesota, United States of America, **2** Institute for Systems Research, University of Maryland, College Park, Maryland, United States of America, **3** Department of Electrical & Computer Engineering, University of Maryland, College Park, Maryland, United States of America, **4** Department of Biology, University of Maryland, College Park, Maryland, United States of America, **5** Robert H. Smith School of Business, University of Maryland, College Park, Maryland, United States of America

* behtash@umd.edu

**Data Availability Statement:** The experimental data used in this paper are publicly available in the Digital Repository at the University of Maryland at http://hdl.handle.net/1903/26351 (at-will attention

## Abstract

Estimating the latent dynamics underlying biological processes is a central problem in computational biology. State-space models with Gaussian statistics are widely used for estimation of such latent dynamics and have been successfully utilized in the analysis of biological data. Gaussian statistics, however, fail to capture several key features of the dynamics of biological processes (e.g., brain dynamics) such as abrupt state changes and exogenous processes that affect the states in a structured fashion. Although Gaussian mixture process noise models have been considered as an alternative to capture such effects, data-driven inference of their parameters is not well-established in the literature. The objective of this paper is to develop efficient algorithms for inferring the parameters of a general class of Gaussian mixture process noise models from noisy and limited observations, and to utilize them in extracting the neural dynamics that underlie auditory processing from magnetoencephalography (MEG) data in a cocktail party setting. We develop an algorithm based on Expectation-Maximization to estimate the process noise parameters from state-space observations. We apply our algorithm to simulated and experimentally-recorded MEG data from auditory experiments in the cocktail party paradigm to estimate the underlying dynamic Temporal Response Functions (TRFs). Our simulation results show that the richer representation of the process noise as a Gaussian mixture significantly improves state estimation and capturing the heterogeneity of the TRF dynamics. Application to MEG data reveals improvements over existing TRF estimation techniques, and provides a reliable alternative to current approaches for probing neural dynamics in a cocktail party scenario, as well as attention decoding in emerging applications such as smart hearing aids. Our proposed methodology provides a framework for efficient inference of Gaussian mixture process noise models, with application to a wide range of biological data with underlying heterogeneous and latent dynamics.

switching experiment) and http://hdl.handle.net/1903/26352 (instructed attention switching experiment).

**Funding:** This research was developed in part with funding from the National Science Foundation (https://www.nsf.gov/) under grant number SMA1734892 awarded to JZS and BB, the National Institutes of Health (https://www.nih.gov/) under grant under grant number R01-DC014085 awarded to JZS, and the Defense Advanced Research Projects Agency (https://www.darpa.mil/) under cooperative agreement number N660011824024 awarded to JZS, MCF, SIM, and BB. The funders had no role in study design, data collection and analysis, decision to publish, or preparation of the manuscript. The views, opinions and/or findings expressed are those of the authors and should not be interpreted as representing the official views or policies of the Department of Defense or the U.S. Government.

**Competing interests:** Sina Miran is employed by Starkey Hearing Technologies, which is a for profit company. The authors declare that no other competing interests exist.

## Author summary

While Gaussian statistics are widely-used in analyzing biological data, they are not able to fully capture the observed heterogeneity and abrupt changes in the dynamics that govern the underlying biological processes. A notable example of such a process is the ability of the human brain to focus attention on one speaker among many in a cocktail party and switch attention to any other at will. We propose a signal processing methodology to extract the dynamics of such switching processes from noisy biological data in a robust and computationally efficient manner, and apply them to experimentally-recoded magnetoencephalography data from the human brain under cocktail party settings. Our results provide new insight on the heterogeneous neural dynamics that govern auditory attention switching. While our proposed methodology can be readily used as a reliable alternative to existing approaches in studying auditory processing in the human brain, it is suitable to be applied to a wide range of biological data with underlying heterogeneous dynamics.

## Introduction

Extracting the latent dynamics that govern biological processes from noisy and limited measurements is a long-standing challenge in computational biology. From the signal processing perspective, state-space modeling is a natural and commonly-used framework for estimation of such latent dynamic processes, i.e., the states, under limited observations [1]. While traditionally used in application domains such as control system design [2], tracking [3], and finance [4], this framework has recently been utilized in the analysis of neural data [5–11]. State-space models (SSMs) often consist of two equations: the state (evolution) equation, to describe the dynamics of the latent process (e.g., the intrinsic level of an internal neural state variable), and the observation equation, to illustrate how the externally-measured observations are related to the process. In signal processing applications, these equations are typically described in a parametric fashion using domain-specific expert knowledge of the problem, and parameter estimation is mostly performed via Expectation Maximization (EM) [12, 13] or Variational Inference (VI) [14, 15]. To better model the state evolution, in addition to expected measurement uncertainties, additive noise terms are often explicitly included in both the state and observation equations. In traditional applications, i.i.d. Gaussian statistics are assumed/imposed on these noise terms to account for the aggregate uncertainties and mismatches in the model. Under linear dynamics and observations, Gaussian noise, and fixed model parameters, Minimum Mean Square Error (MMSE) state estimation is conducted by the well-known Kalman filter and smoother [1]. For more general SSMs, Sequential Monte Carlo (SMC) methods can be used for MMSE state estimation [16]. In the context of neuroimaging data analysis, SMC methods have been utilized in MEG dipole modeling and source localization [7–9].

Gaussian statistics, however, are often inconsistent with the empirical histograms of the observations in various applications, including neuroimaging data analysis. For instance, in MEG analysis, the observation noise consists of intrinsic magnetic noise, ocular or motion-induced artifacts, as well as background activity unrelated to the stimulus. While the intrinsic noise can be reliably modeled by Gaussian statistics and estimated from stimulus-free measurements in experimental settings, the artifacts and neural background activity are manifestly non-Gaussian and non-stationary. However, when there is direct access to the observed signals, source separation techniques have been successfully utilized to remove and mitigate these latter sources of uncertainty [17–22].

Similarly, the state model noise terms introduced above, often referred to as process noise, do not actually follow Gaussian statistics in various real-world applications [23, 24]. This is mainly due to two reasons: First, in time series analysis, abrupt state changes may not be well represented by Gaussian statistics. Second, in practice, the statistics of the process noise heavily depend on the specifics of the experimental design, such as the task demand and subject's performance, as well as other exogenous variables not accounted for. Critically, unlike the case of the observations, states are only indirectly observed, which limits the utility of source separation techniques. Finally, despite the negative connotation of the word "noise", the process noise also captures the model-critical stochasticity of the state evolution. As such, the goal is to model and account for said stochasticity, as opposed to removing it as in the case of observation noise.

This issue is particularly important in modeling brain function as a latent dynamic process: taking the states to represent the underlying neural circuits that process sensory stimuli, the process noise then consists of both the underlying behaviorally- and stimulus-driven dynamics as well as the background neural activity (not necessarily evoked by the stimulus or behavior), which are typically quite structured and far from being Gaussian. In this context, the state evolution model is more prone to model mismatch and biases, as compared to the observation equation, considering that we generally have more control over the measurement system than the generative mechanism governing the latent process. As a result, the empirical histogram of the process noise (which can be computed from state estimates) could exhibit multimodal morphology, with each mode corresponding to a different exogenous process driving the state dynamics during specific portions of the experiment.

This has led researchers to study SSMs with a Gaussian Mixture (GM) process noise [25–29] considering that a GM can, in principle, approximate any multimodal density [30]. These existing results primarily focus on state estimation and approximation of filtering and smoothing densities under a *fixed* or *known* GM noise density. As such, parameter estimation for a GM process noise in SSMs has not been well-studied. Switching SSMs has been another direction of research in extending linear Gaussian SSMs to cope with nonstationarity, model mismatch, and exogenous processes [15, 31–39]. In this approach, several linear Gaussian SSMs are considered to underlie the observed time series data, which switch according to a Hidden Markov Model (HMM). Although the filtering and smoothing densities in this model take a GM form, the potential multimodality of the process noise is not explored or modeled in this approach. In addition, parameter estimation for switching SSMs is a challenging task in general, due to the intricate dependence of the data likelihood on the parameters. When the states are directly observable, the resulting models are known as Markov-switching Autoregressive (MSAR) models, which notably admit parameter estimation via the EM algorithm [32, 40]. However, for general switching SSMs, parameter estimation often requires computationally intensive numerical optimization steps [33, 35, 36].

In this work, we fill this gap by developing an EM-based algorithm for estimating the parameters of a GM process noise model from the observations in an SSM. In our model, the process noise is not drawn i.i.d. from a GM but instead, a GM component is chosen at random for a window of fixed (but arbitrary) length, and the process noise within the window is drawn from said component. The parameters of the GM are unknown. The EM algorithm has been widely used for parameter estimation both in state-space modeling [13] and in GM clustering [41], which makes it a promising candidate for our setting. The EM framework in this setting, however, results in intractable expectations for parameter updates. We address this issue by leveraging a Sequential Monte Carlo Expectation Maximization (SMCEM)-type algorithm [42] to approximate the expectations using smoothed particles obtained through SMC. A major drawback of particle smoothing approaches is their excessive computational

requirements, or equivalently suffering from sample depletion as the dimension of the target densities grows while fixing the computational costs [43]. As a more scalable alternative, we develop another method of approximating the expectations based on closed-form approximations to the smoothing densities as well as their one-step cross covariances. To this end, we adopt the two-filter formula for smoothing [26] and devise a belief propagation algorithm in our setting. As a result, the computational complexity of the E-step in EM for a GM process noise would be comparable to that of a conventional Gaussian process noise, akin to performing parallel Kalman filtering and smoothing.

To demonstrate the benefits of a GM process noise and the efficacy of the developed estimation framework, we consider two experimental paradigms: a dynamic at-will attention switching task in a realistic cocktail party scenario, in which the listener maintains attention to one out of two competing speakers, while being able to switch attention between the two speech streams at will; and an instructed attention switching task in a more restricted cocktail party scenario, in which the listener maintains attention to one out of two competing speakers for the first half of a trial and then switches attention to the other speaker. The cocktail party is among the key paradigms in studying the neural dynamics underlying complex auditory processing [44, 45]. One of the most recent quantitative approaches in uncovering these neural dynamics from neuroimaging data is based on the Temporal Response Function (TRF) model [46]. The TRF can be considered as an evolving Finite Impulse Response (FIR) filter which gets convolved with speech features in time, e.g., the speech envelope, to produce the auditory neural response observed through neuroimaging modalities such as electroencephalography (EEG) and magnetoencephalography (MEG) [47]. The TRF framework has resulted in new insights into the mechanisms of speech processing in the brain in the cocktail party scenario [45, 48–50]. For instance, TRF components at specific lags may exhibit peaks which *arise*, *persist*, and *disappear* over time according to the attentional state of the listener [51]. The different local dynamics of TRF components under each of these conditions motivates a GM density to capture such evolution patterns. Dynamic estimation of TRFs was first discussed in [47] using a Recursive Least Square (RLS) algorithm. However, smoothing estimates and state-space modeling are more robust than RLS and filtering estimates in performing a comprehensive dynamic analysis of TRFs when data from multiple trials is available. Thus, we study dynamic estimation of TRFs using SSMs and apply our SSM framework with a GM process noise to both simulated and experimentally recorded MEG data under a dual-speaker environment where the subject switches attention between the two speakers at will. The results show that our proposed algorithm can effectively recover the multimodal structure of the process noise from SSM observations, and that having a richer and more realistic representation of the process noise allows capturing the TRF dynamics more precisely and more consistent with the subjects' behavioral reports, as compared to the conventional Gaussian SSM or RLS estimation. While our proposed framework is motivated by and applied to data from auditory experiments, it is applicable to general state-space modeling problems in which states exhibit heterogeneous and recurring local dynamic patterns.

## Results

In this section, we demonstrate the utility of our proposed algorithms in estimating TRFs from auditory neural responses to speech, using both simulated and experimentally-recorded MEG data. Before doing so, we will give an overview of the TRF model, existing estimation frameworks, and the benefits of our GM SSM framework for TRF estimation.

## The TRF model

Consider a cocktail party setting [45], in which a subject is listening to two speakers simultaneously, but only attending to one of the speakers. While the subject is performing this task, the neural response is recorded using MEG. The TRF is a commonly used linear encoding model that relates the speech features to the neural response, by generalizing the concept of event-related evoked responses: instead of averaging over multiple trials with the same stimulus to obtain the evoked response, the TRF kernel is obtained by averaging the effect of a diverse set of speech stimuli, presented as a continuous time series, and hence results in a generalizable encoding model (See Fig 1 for a schematic depiction). The speech features used in TRF models have included the acoustic envelope, acoustic onsets, phoneme representations, word frequency measures, and semantic composition [52–54]. In a multi-speaker scenario, multiple TRFs are used to capture the effect of the speech features of each speaker to the neural response.

Existing results in auditory neuroscience [11, 46–48, 51, 55] have focused on studying the behavioral significance of the various peaks in the TRF. For instance, the TRF exhibits an early positive peak at around 50 ms, referred to as the M50 component, which is known to represent the encoding of the acoustic envelope. A later negative peak at around 100 ms lag, referred to as the M100 component, has shown to have an attentional modulation effect, so that it appears to have a larger magnitude for the attended speaker's TRF, compared to the unattended speaker's TRF. The M50 component is attributed to the effect of early auditory processing in the brain and is equally represented in both speakers' TRFs, while the M100 component represents the later processing stages that segregate the attended speaker from the unattended one [48].

TRFs are commonly assumed to be static during the duration of an auditory experiment, and are estimated using regularized least squares [56, 57] or boosting [46, 48, 58]. Dynamic estimation of the TRFs, on the other hand, can provide insights into the underlying neural dynamics that process speech in the cocktail party setting, and has significant implications for the design of non-invasive brain-machine interface devices involving auditory processing, such as the emerging 'smart' hearing aid technology that utilizes neural signals to steer the hearing aid parameters in real-time.

Dynamic estimation of TRFs was first discussed using a regularized RLS framework in [47]. This method considers changes in the TRFs over consecutive non-overlapping time windows

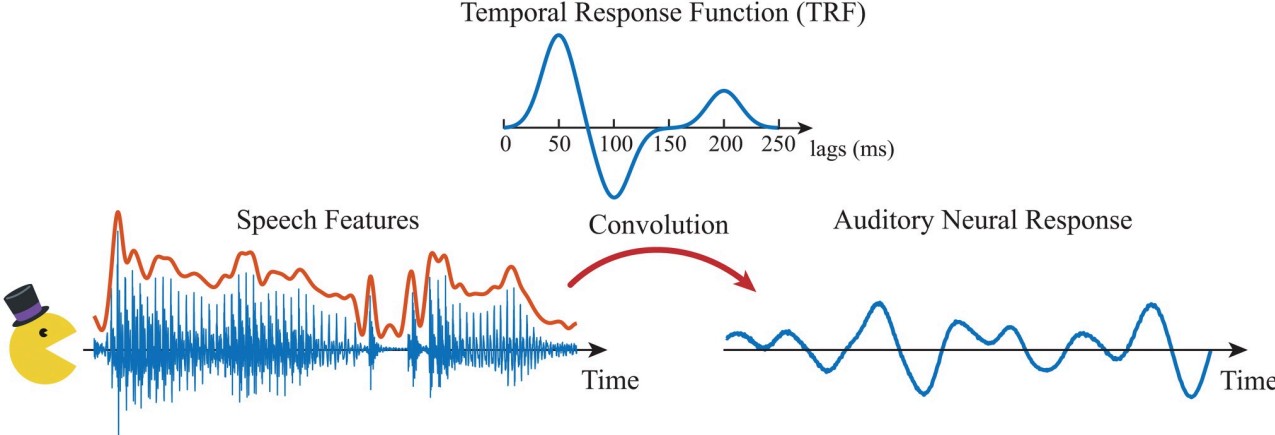

**Fig 1.** Schematic depiction of the TRF model. The speech features (left, e.g., acoustic envelope) are convolved with the TRF (top) to predict the auditory neural response (right).

of small length, and updates the estimates of the TRFs in a recursive fashion as more data becomes available (See Methods for more details). As such, it provides filtered estimates of the TRFs and is suited for real-time applications.

Leveraging SSMs for representation and estimation of the TRFs has the advantage of providing smoothed estimates and directly modeling the evolution of the TRFs through the state equation, and thereby resulting in a more precise dynamic analysis of the TRFs in the off-line fashion. In this work, we consider linear Gaussian SSMs and linear SSMs with GM process noise. As we will demonstrate in the following two subsections, the linear SSMs with GM process noise have the additional advantage of accounting for the heterogeneity of the TRF dynamics.

In what follows, we consider regularized RLS estimates of the TRF, estimates of the TRF using a Markov Switching Autoregressive (MSAR) model, and smoothed TRF estimates from a linear Gaussian SSM as benchmarks (See Methods for more details on these algorithms).

## Application to simulated data

Consider a 90 s long cocktail party experiment, in which the subject is listening to two speakers simultaneously and is instructed to switch attention between the two every 15 s starting at time 7.5 s. We synthesize the putative TRF dynamics as shown in Fig 2A, based on the relevance of different TRF peaks. We use a sampling rate of $F_s$ = 100 Hz and a length of 250 ms for the TRFs. The TRFs are represented using a dictionary with five Gaussian atoms with variances of 0.018 $s^2$ whose means are separated by 50 ms increments starting from a lag of 0 ms to 200 ms. Furthermore, we consider a piecewise-constant model for the TRFs over windows of length 300 ms. Letting $\tilde{\mathbf{G}}$ be the dictionary, the TRF at the $n^{\text{th}}$ window is defined as $\boldsymbol{\tau}_n = \tilde{\mathbf{G}}\mathbf{x}_n$, where $\mathbf{x}_n$ is the *state* vector at window $n$. The SSM governing the state evolution is of the form $\mathbf{x}_n = \alpha\,\mathbf{x}_{n-1} + \mathbf{w}_n$, where $\alpha < 1$ is a constant and $\mathbf{w}_n$ is the process noise. Finally, the observed neural response is related to the states by $\mathbf{y}_n = \mathbf{S}_n^{\top}\boldsymbol{\tau}_n + \mathbf{v}_n$, where $\mathbf{S}_n$ are the speech features of the two speakers relevant to window $n$ and $\mathbf{v}_n$ is the i.i.d. Gaussian observation noise, i.e., $\mathbf{v}_n \sim \mathcal{N}(\mathbf{0}, \sigma^2\mathbf{I})$ (See Methods for more details on the TRF and state-space models).

Fig 2A shows the synthesized TRF heatmaps for speakers 1 and 2, where the corresponding states are designed such that the M50 component stays relatively constant for the two speakers, the M100 component is modulated by the attentional state, and a common high-latency component at 200 ms varies independently of the subject's attention. Fig 2B shows two snapshots of the TRF of speaker 2 at 10 s, when speaker 2 is attended, and at 85 s, when speaker 1 is attended. It is worth noting that the corresponding states in Fig 2A are not generated from an SSM. However, the relatively smooth temporal changes of the TRFs in Fig 2A (representing neural activity in controlled experimental conditions) makes the SSM model a suitable candidate for dynamic TRF analysis. Indeed, the TRF components at lags of 100 ms and 200 ms exhibit heterogeneous dynamics across the trial, including periods of increasing, decreasing, and remaining relatively constant, which model the changes in auditory state throughout the experiment. Such dynamics can be modeled using a multimodal process noise $\mathbf{w}_n$. Fig 2C shows the histogram of true $\mathbf{w}_n$ samples along with the 3rd state dimension of speaker 2's TRF (corresponding to the M100 component). The true process noise samples are computed as $\hat{\mathbf{w}}_n = \mathbf{x}_n - \alpha\mathbf{x}_{n-1}$, assuming that the true states ($\mathbf{x}_n$'s) in Fig 2A are available to an oracle. We refer to this histogram as the oracle histogram and to the maximum-likelihood GM density fit to these oracle samples as the oracle GM fit in Fig 2C. The constant $\alpha$ is chosen close to and less than one to enforce temporal continuity. We assume that the TRF dynamics are governed by one mixture component in each window of length 1.5 s. We simulate the observed neural

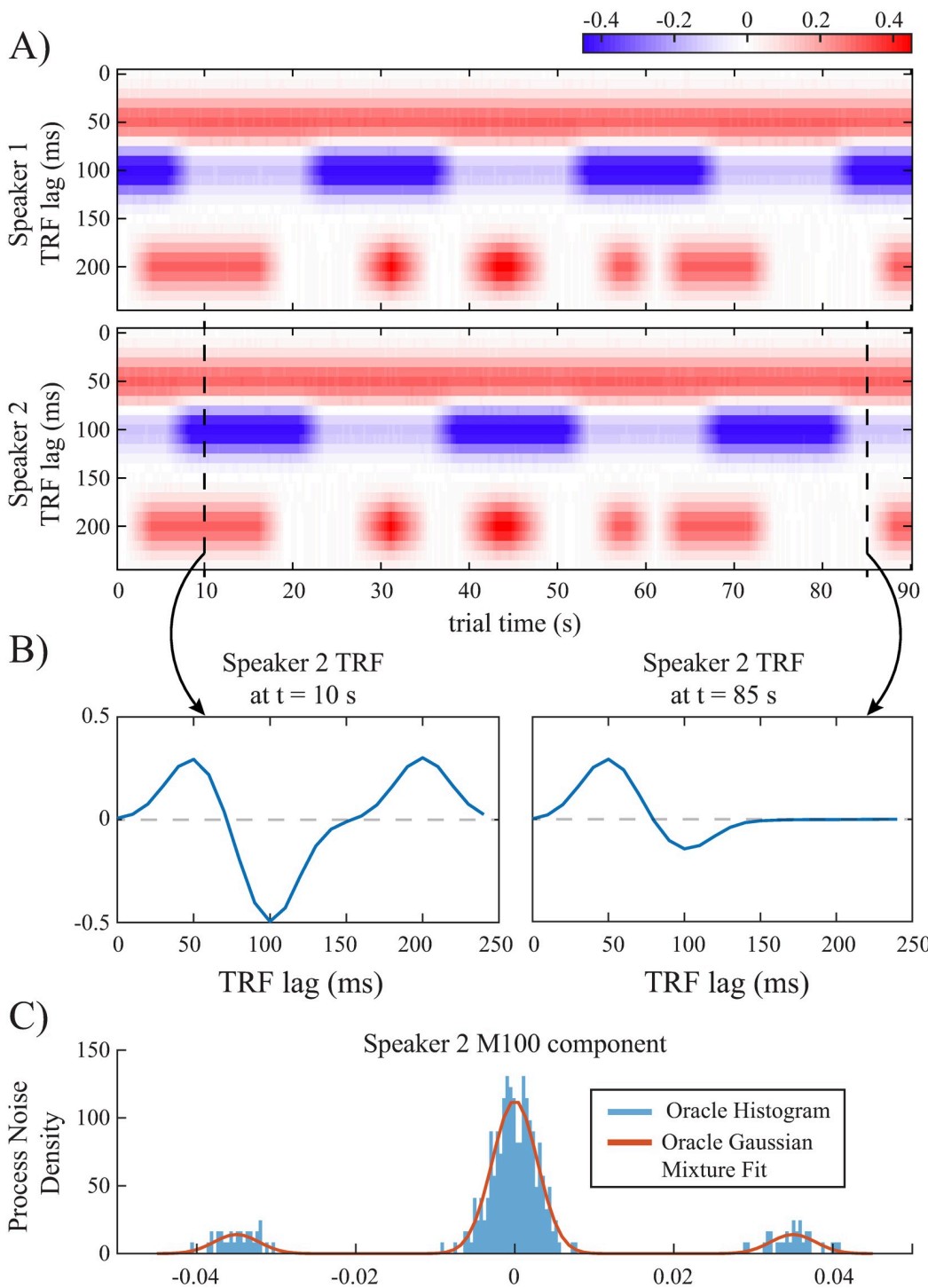

**Fig 2.** Designed simulation study: A) Heatmaps of the synthetic TRFs in time for a two-speaker cocktail party scenario, where the M100 magnitudes are attention-modulated. B) Example instances of speaker 2's TRF when the speaker is attended (left plane) and unattended (right plane). C) Oracle histogram of process noise in (14) along the M100 dimension of speaker 2, which is computed from (A), and the fitted GM as the oracle GM fit.

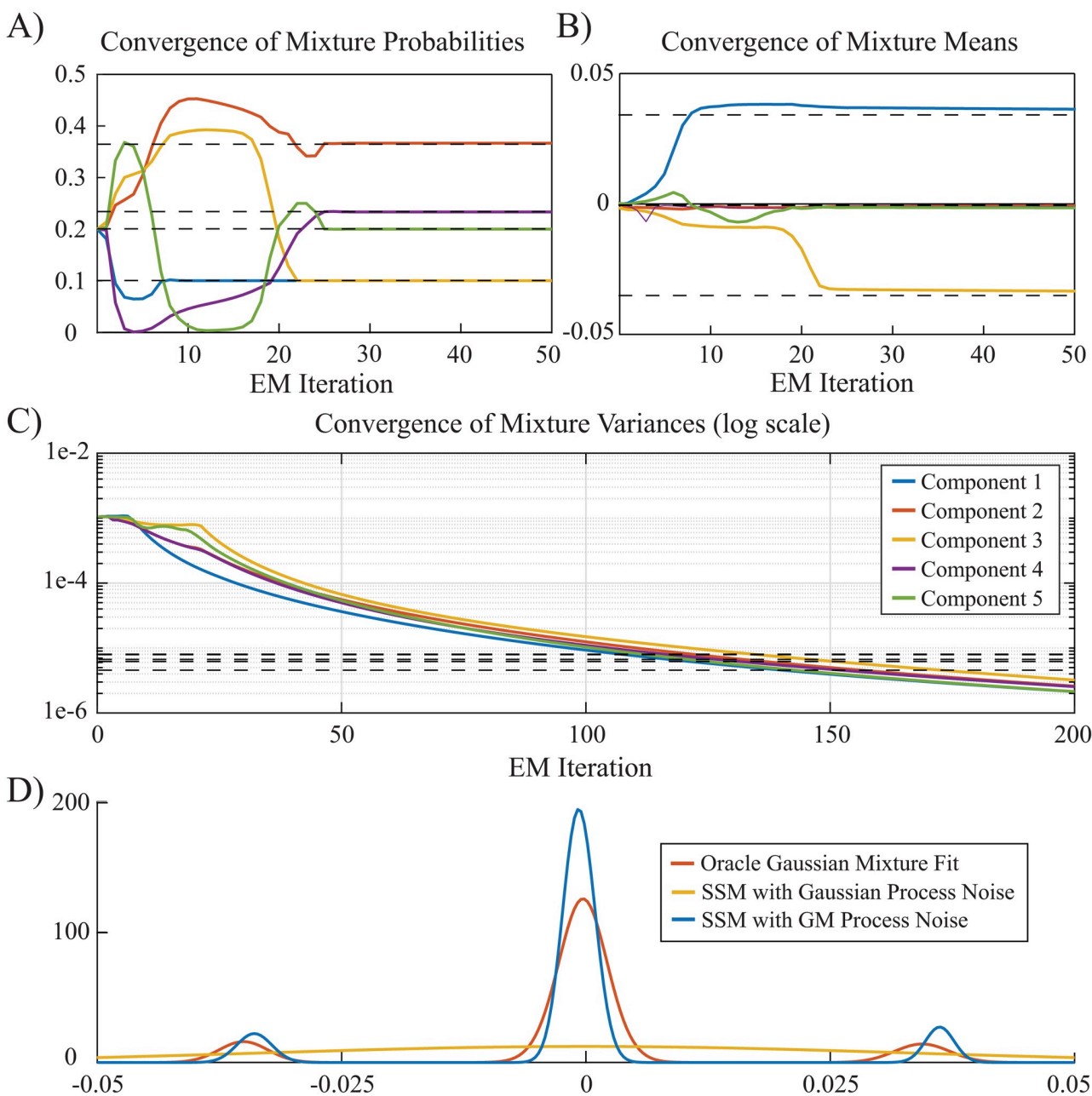

**Fig 3.** Convergence of Gaussian mixture parameters for $M = 5$ in the EM algorithm with closed-form approximations: A) Mixture probabilities. B) Mixture means (along the M100 component of speaker 2 as an example). C) Mixture variances (along the M100 component of speaker 2 as an example). Bold dash lines show the corresponding parameters of the oracle GM fit. D) GM densities (along the M100 component of speaker 2 as an example).

response $y_t$ using two speech signal envelopes as the stimulus vectors (See Methods for more details on the parameter settings).

Fig 3 shows the convergence of the estimated parameters using the proposed EM algorithm in comparison to those given by the oracle GM fit for a nominal observation SNR of 6.7 dB, using the closed-form approximation approach. The number of mixture components is chosen as 5 using the Akaike Information Criterion (AIC). The observation noise variance $\sigma^2$ is also

estimated within the EM algorithm. The panels for the means and diagonal covariances in Fig 3 correspond to the 3rd state dimension of speaker 2's TRF (i.e., the M100 component) from Fig 2C. The mixture probabilities and means of the oracle GM fit are recovered within 30 EM iterations. The covariance elements, however, take more iterations to converge and tend to underestimate those of the oracle GM fit. This shows that at the nominal SNR of 6.7 dB in our simulation, the algorithm is more sensitive to recovering the average TRF dynamics in each 1.5 s window than to retrieving the detailed variations within the window.

It is noteworthy that the initialization points in Fig 3C, given by the estimated process noise variance in a Gaussian SSM, are approximately 100 times larger than those given by the oracle GM fit. Fig 3D shows the corresponding estimated process noise density after 200 EM iterations (blue trace), the oracle GM fit (red trace), and the Gaussian model fit obtained from a linear Gaussian SSM used for EM initialization (yellow trace). While the estimated GM process noise density using our proposed approach closely matches that given by the oracle GM fit, the process noise density obtained by a linear Gaussian model is heavily biased and is not able to capture the multimodal nature of the process. Note that while Fig 2C alludes to a true density with 3 GM components, the AIC criterion chose 5 GM components. Nevertheless, the joint updating of the means, variances, and mixture components (Fig 3A, 3B and 3C) results in a final density estimate that matches the putative true density with 3 GM components (Fig 3D). As such, our algorithm exhibits robustness to overestimation of the number of mixture components.

To ease reproducibility, we have archived a MATLAB implementation of the closed-form approximation method in the GitHub repository, which reproduces the results of Fig 3 [59]. Convergence curves for the Monte Carlo approximation method are previously presented in [60], and are omitted here for brevity.

Fig 4 shows the normalized RMSE in state estimation with respect to the original states in Fig 2A for nominal observation SNRs in the range [-5.3, 9.7] dB with 3 dB increments. The results are averaged over 10 realizations at each SNR value. The SSMs clearly outperform the RLS and MSAR algorithms in recovering the true states. Also, the SSM with GM process noise with either the closed-form or particle smoothing approximations outperforms the Gaussian SSM. We have considered a total of 2000 particles for the particle smoothing algorithm (Approach 1) so that state estimates are comparable to those obtained by the closed-form approximation (Approach 2). This resulted in a ten-fold increase in the run-time compared to the closed-form approximation method (61.50 seconds and 5.57 seconds for Approaches 1 and 2, respectively, per EM iteration, on a typical desktop workstation for the settings used in the simulation), which shows the advantage of using the closed-form approximation method. Examples of the estimated TRFs of speaker 1 under the low nominal observation SNR of -5.3 dB are shown in Fig 5. The MSAR (panel B) and RLS estimates (panel C) exhibit the highest variability compared to the ground truth in Fig 5A (imported from Fig 2A). While the Gaussian SSM estimate in Fig 5D fails to capture the rapid M100 dynamics as well as the steady M50 component (note the M50 and M100 estimates within the dashed rectangles), the estimate from the SSM with GM process noise in Fig 5E is nearly indistinguishable from the ground truth TRF in Fig 5A.

## Application to experimentally-recorded MEG data

We present the analysis of data from two separate attention switching experiments, which we refer to as the at-will and instructed attention switching experiments. In the at-will attention switching experiment, subjects listened to a speech mixture, and were instructed to start attending to the male speaker first, and then to switch their attention between the two speakers

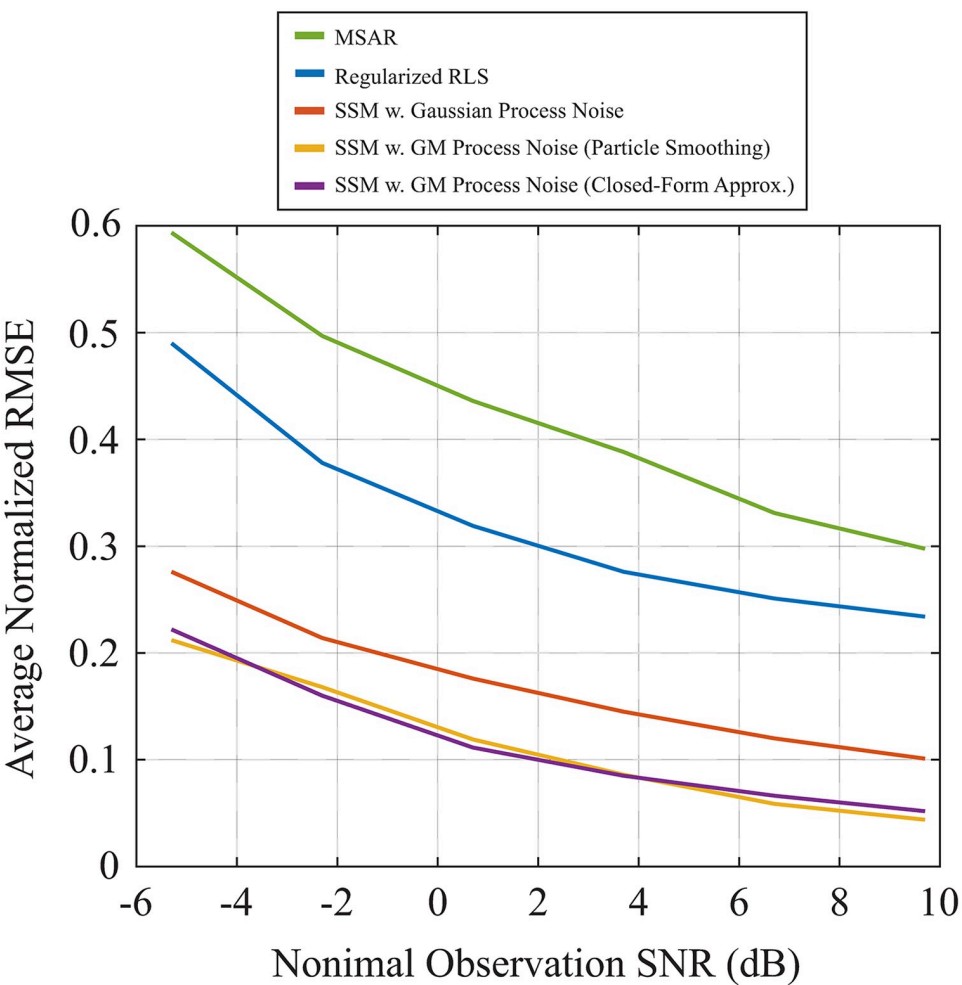

**Fig 4.** Averaged normalized RMSE in state estimation computed over 10 runs of observation noise at each SNR value for dynamic TRF estimation algorithms, namely, MSAR, regularized RLS, linear Gaussian SSM, and linear SSM with GM process noise using closed-form and Monte Carlo particle smoothing approximations. States and noise parameters are both estimated simultaneously from the observations in each run.

at their own will for a minimum of one and a maximum of three times during each trial. In the instructed attention switching experiments, the subjects listened to a speech mixture, and were instructed to start attending to one speaker first and then to switch their attention to the other speaker halfway through the trial. The instructed attention switching experiment consists of data from 7 subjects, with 6 trials each, while the data from the at-will attention switching experiment pertains to 3 trials of one subject (See Subjects, Stimuli, and Procedures subsection in the Methods for more details). Although reliable group-level conclusions for the challenging at-will attention switching experiment require data from more subjects and trials, given the novelty of this experimental paradigm and its potential interest to the auditory attention decoding research community, we have included the analysis of data from this one subject, separately from the instructed attention switching data. In addition, a GM process noise in SSMs would be more beneficial in at-will attention switching experiments, as it can capture the rapid dynamics that underlie attention switching instances more reliably.

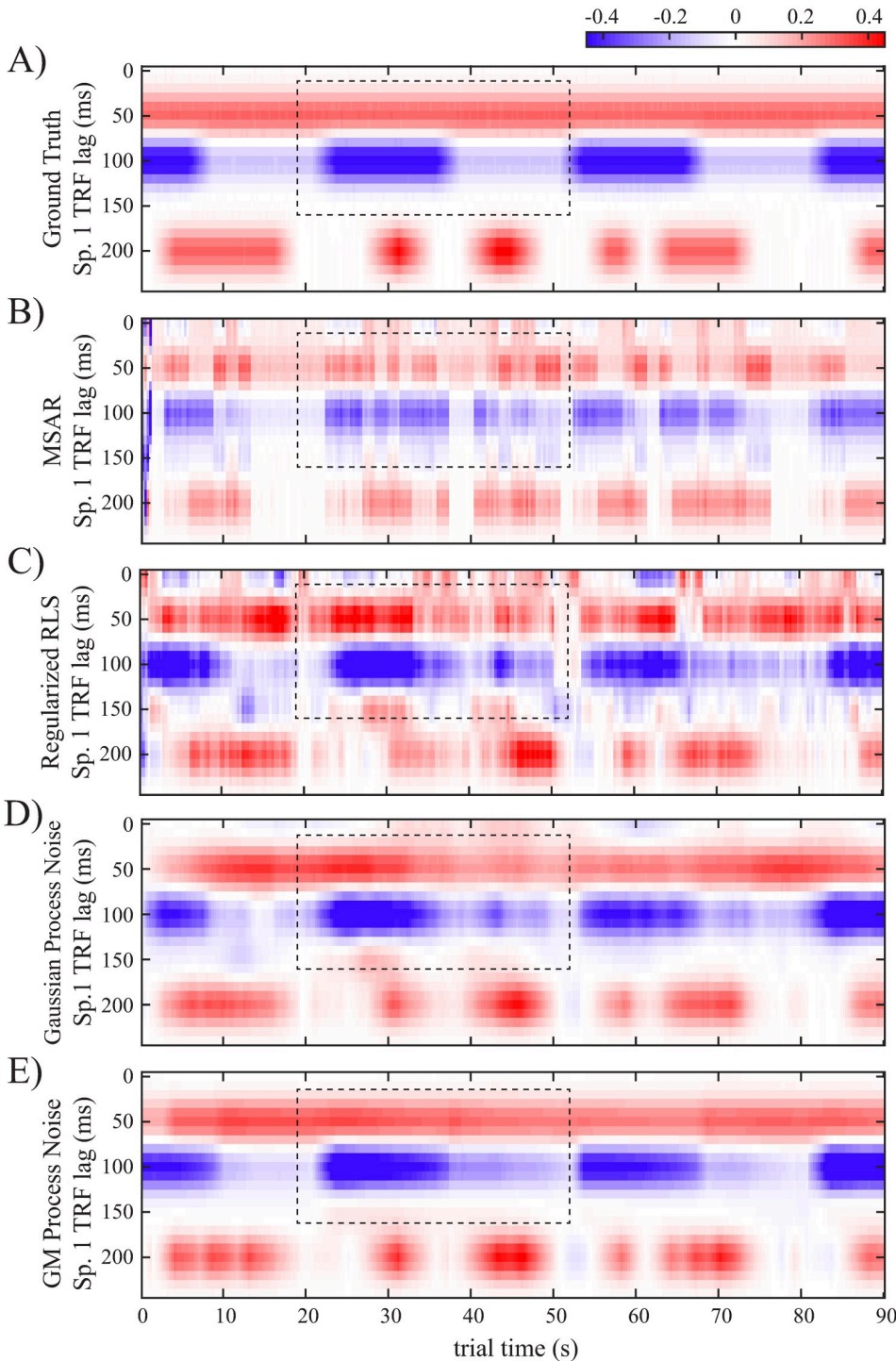

**Fig 5.** Example dynamic TRF estimates for speaker 1 under the low nominal observation SNR of −5.3 dB: A) The ground truth TRF. B) MSAR. C) Regularized RLS. D) Linear Gaussian SSM. E) Linear SSM with GM process noise. The dashed rectangles highlight an example difference of these estimates for the sake of comparison.

**TRF estimation results and discussion.** We set the TRF length to 300 ms and consider TRFs to be piece-wise constant over windows of length 400 ms. Also, we assume that the TRF dynamics are governed by one mixture component in each window of length 2 s. As before, we represent the TRFs over a Gaussian dictionary with means separated by 20 ms starting from 0 to 280 ms, and variances of $8.5 \times 10^{-3}$ s$^2$. To restrict the dynamic range of the process noise $\mathbf{w}_n$ for the sake of robustness, we consider Inverse Gamma (IG) conjugate priors [61] on the diagonal elements of the process noise covariance matrices. Note that in the absence of such priors, the EM algorithm would likely result in TRFs that are highly variable in time and with no meaningful morphological structure (See Methods for more details on parameter settings).

Fig 6 shows example TRF estimates for two trials of the subject in the at-will attention switching experiment. The vertical dashed lines mark reported attention switches by the subject. For the sake of brevity, hereafter we only present TRF estimates based on the RLS, Linear Gaussian SSM and Linear SSM with GM process noise. The number of mixture components for the process noise was set to 3 for trial 1 and 4 for trial 2, using the AIC criterion. Row A shows speaker 1's TRF estimate using RLS, which exhibits the highest variability. Rows B and C show the TRF for the Gaussian SSM and the SSM with GM process noise, inferred using the closed-form approximation, respectively. Although the estimated process noise variance in the GM case is controlled by that of the Gaussian case in each dimension, we observe that the estimates in row C clearly delineate the heterogeneity of the dynamics of the various TRF components, which are blurred by the linear Gaussian SSM estimates of row B. In other words, the multimodal representation of the process noise allows the model to adapt to rapid changes governed by the subjects' behavior. Row D displays speaker 2's TRF estimate using the linear SSM with GM process noise. Comparing rows C and D, we observe the aforementioned attention modulation effect in the magnitude of the M100 components. To illustrate this effect further, row E shows the difference between the M100 magnitudes of the TRFs of speakers 1 and 2, where we locate the M100 at each time as the smallest TRF elements in the [0.1, 0.2] s lag interval. Thus, when speaker 1 (2) is attended, we expect this difference to be positive (negative). The attention decoding accuracy in each trial can be computed by comparing the difference of the M100 magnitudes with level 0 at each time (horizontal dashed line in Fig 6E) considering the reported attended speaker and summing over all the intervals where the M100 of the attended speaker exhibits a larger magnitude than that of the unattended speaker. Note that this decoding strategy is purely based on the TRF estimates in a single trial. As such, it would not be as accurate as the state-of-the-art attention decoding methods that use more complex algorithms and extensive training data. The M100 differences for the RLS exhibit high variability (blue traces), and result in inconsistencies with the reported attended speakers (e.g., trial 1 after the 35 s mark, downward arrow). The M100 differences obtained by the linear Gaussian SSM estimates seem to overly smooth those of the RLS (e.g., trial 2, near the 10 s mark, downward arrow). The M100 differences obtained from the proposed linear SSM with GM process noise, however, provide a desirable compromise between these two extremes: Compared to the linear Gaussian SSM, the M100 differences benefit from the clearly delineated TRF dynamics and can result in earlier detection of an attention switch, leading to higher attention decoding accuracy. Instances of this advantage are marked by green arrows in row E, for both trials. Decoding the attention based on the sign of the M100 differences results in misclassification rates of (6.74%, 12.37%, 22.94%) for trial 1 and (31.72%, 43.08%, 39.03%) for trial 2, respectively for the SSM with GM process noise, Gaussian SSM, and RLS, in accordance with the foregoing qualitative analysis.

Fig 7 displays example TRF estimates for two trials in the instructed attention switching experiment, in a similar fashion as Fig 6. The subjects were instructed to switch their attention

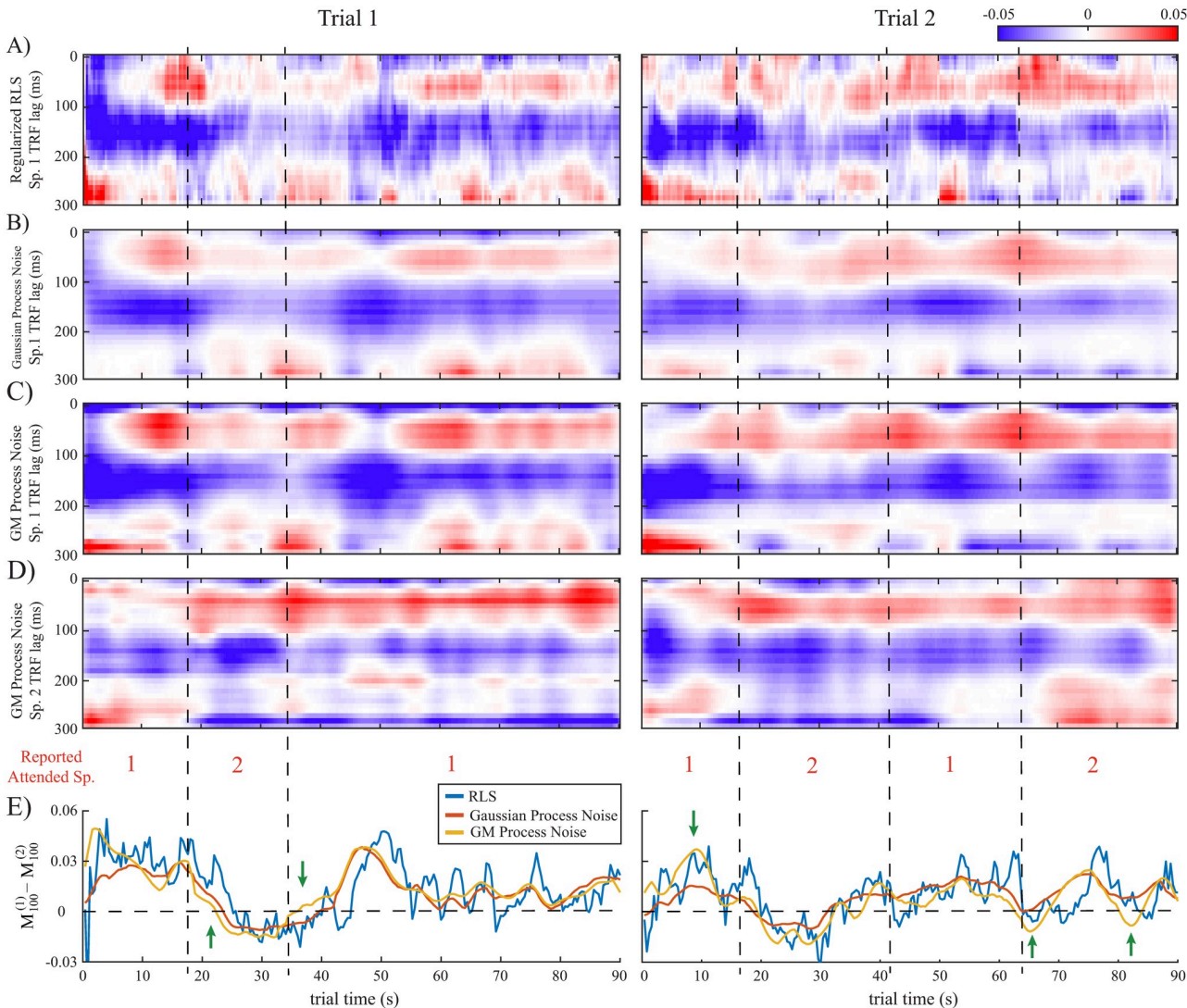

**Fig 6.** TRF estimates for two example trials in the at-will attention switching experiment with vertical dashed lines showing the reported times of attention switches by the subject: A) RLS estimate (speaker 1 TRF). B) Gaussian SSM (speaker 1 TRF). C) SSM with GM process noise (speaker 1 TRF). D) SSM with GM process noise (speaker 2 TRF). E) M100 magnitude differences between the TRFs of speaker 1 and 2 for the different methods. The SSM with GM process noise better delineates the heterogeneity of the TRF dynamics and is more consistent with the subjects' behavioral reports (see green arrows), while the RLS estimate is highly variable and the estimate of the Gaussian SSM is overly smooth.

at the 30 s mark, halfway through the trial. Again, we observe that the SSM with GM process noise emphasizes the detailed dynamics of the TRFs which are sometimes blurred out in the Gaussian SSM or shown with high variability in the filtering estimates of RLS. This can result in stronger attention modulation effects, i.e., larger magnitude for the M100 of the attended speaker, or quicker transitions at the 30 s attention switching mark (marked by green arrows). The misclassification rates for SSM with GM process noise, Gaussian SSM, and RLS are respectively (25.78%, 26.81%, 31.45%) for trial 1 and (8.55%, 9.8%, 17.01%) for trial 2.

Fig 8 shows the group-level analysis results for the subject in the at-will attention switching experiment (left column) and the seven subjects in the instructed attention switching experiment (right column). The upper panels display the scatter and box plots for the computed attention decoding accuracies for the at-will and instructed attention switching experiments,

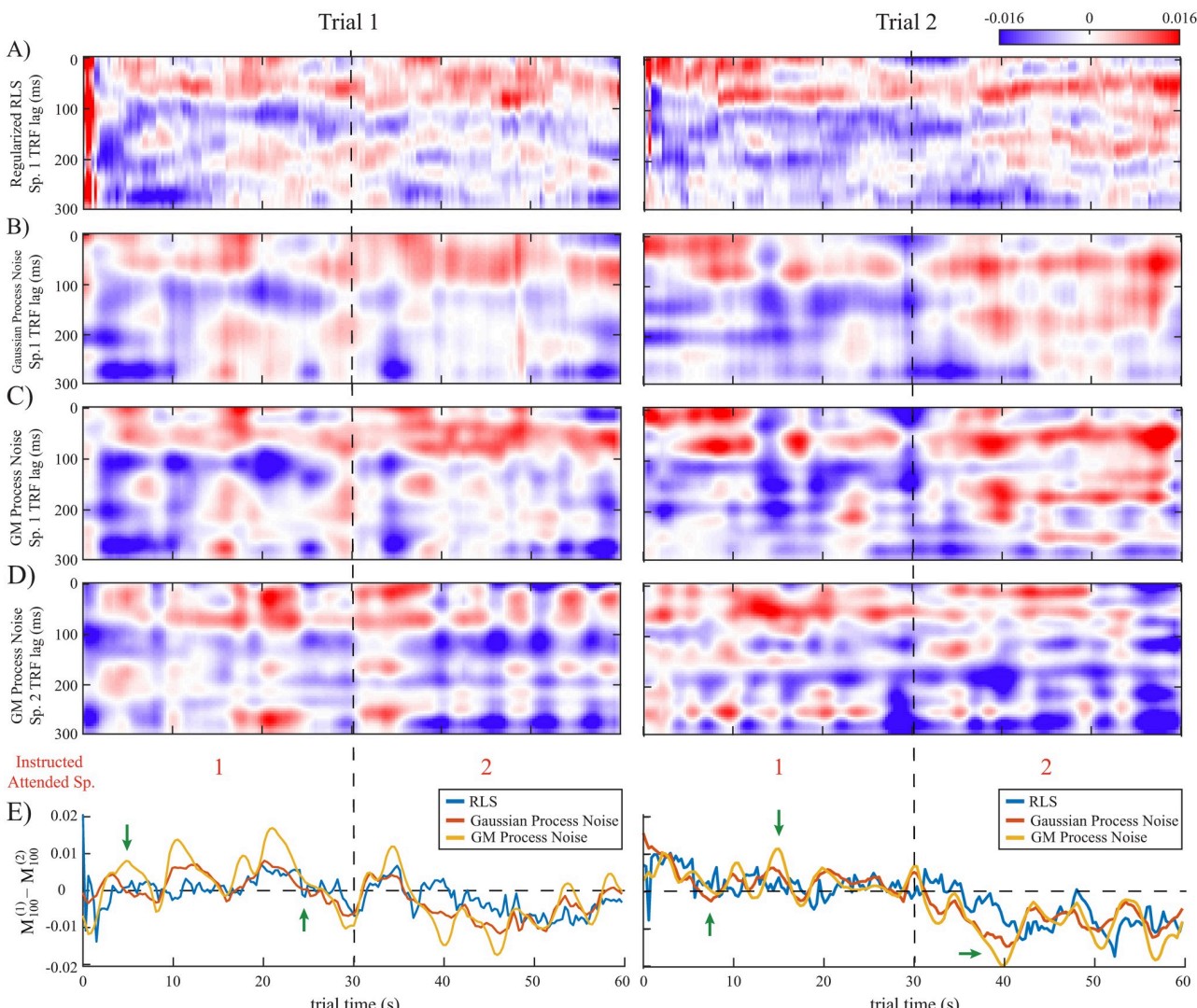

**Fig 7.** TRF estimates for two example trials in the instructed attention switching experiment with vertical dashed lines showing the 30 s mark where subjects were instructed to switch their attentional focus: A) RLS estimate (speaker 1 TRF). B) Gaussian SSM (speaker 1 TRF). C) SSM with GM process noise (speaker 1 TRF). D) SSM with GM process noise (speaker 2 TRF). E) M100 magnitude differences between the TRFs of speaker 1 and 2 for the different methods. The SSM with GM process noise better delineates the heterogeneity of the TRF dynamics and is more consistent with the subjects' behavioral reports (see green arrows), while the RLS estimate is highly variable and the estimate of the Gaussian SSM is overly smooth.

respectively. With respect to the mean attention decoding accuracy (red plus sign), RLS estimates exhibit the poorest performance, and SSM with GM process noise results in a modest 1% to 3% improvement over the linear Gaussian SSM. The attention decoding accuracy, however, is not an ideal metric to assess the TRF estimation performance, due to the absence of a ground truth and arbitrary attentional variabilities during a single trial. The lower panels in Fig 8 summarize the results of the AIC model selection criterion for number of components in the GM process noise density. For each trial, we considered one to four GM components for the process noise, and the number with the lowest AIC score was chosen for the SSM with GM process noise in that trial. If one GM component is chosen based on the AIC criterion, the SSM with GM process noise reduces to the linear Gaussian SSM. The lower panels in Fig 8 show the normalized histogram of the chosen number of GM components across subjects and

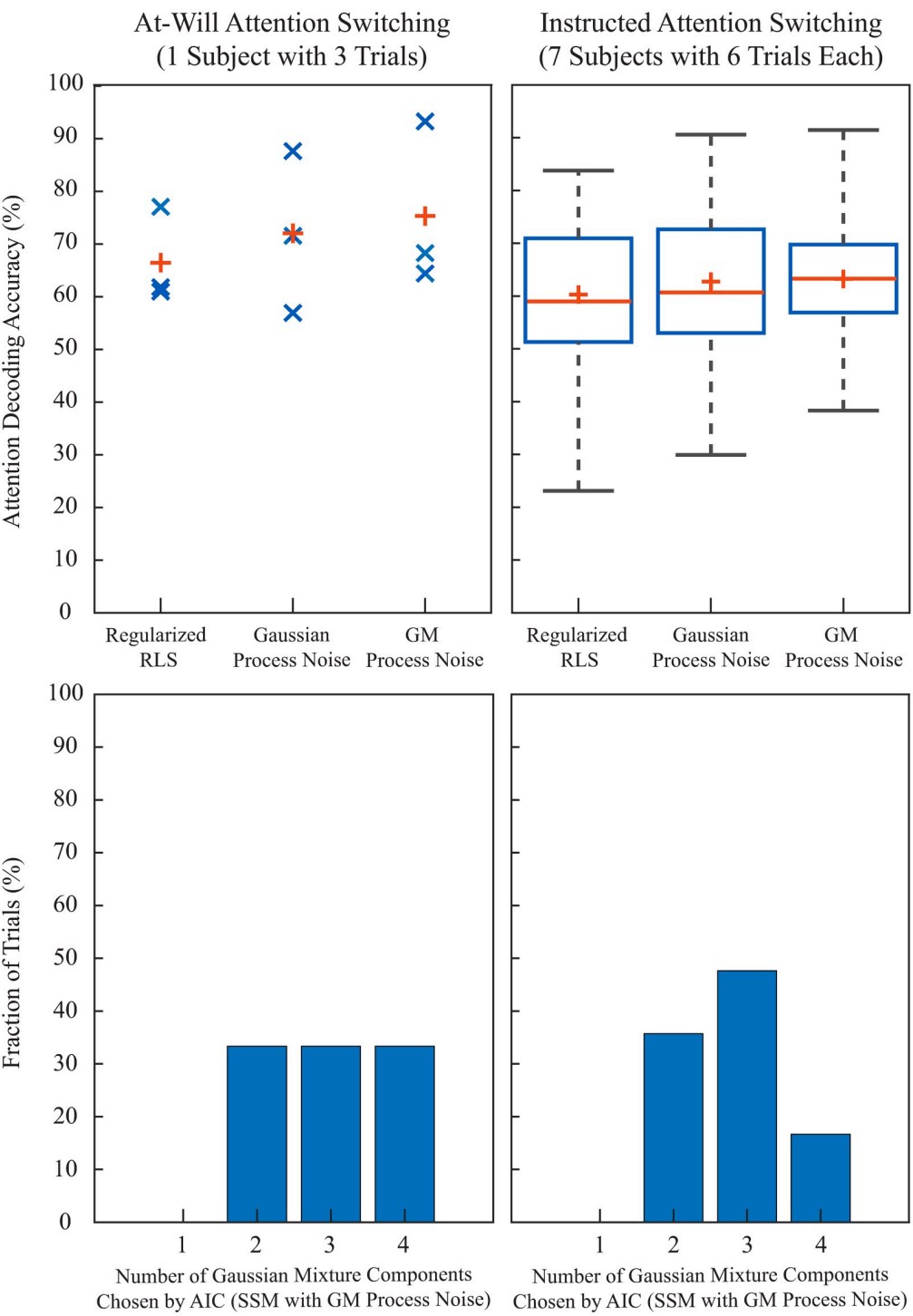

**Fig 8.** Group-level results for attention decoding accuracy (upper panels) and the AIC model selection criteria (lower panels) for the at-will (left column) and the instructed (right column) attention decoding experiments. The upper panels display the scatter/box plots of trial-level mean attention decoding accuracy across subjects and trials, while the lower panels show the normalized histogram of the chosen number of GM components in each trial. The mean attention decoding accuracies, marked by red plus signs in upper panels, are (75.31%, 72.04%, 66.45%) and (63.24%, 62.74%, 60.28%) for (SSM with GM process noise, Gaussian SSM, RLS) in the at-will and instructed attention switching experiments, respectively. Although the SSM with GM process noise results in modest improvements in mean attention decoding accuracy compared to the linear Gaussian SSM, the AIC model selection criteria always prefers the former to the latter.

trials. It is worth noting that for the trials where four components were chosen, it is possible that a higher number of process noise GM components would have resulted in a lower AIC score. We observe that in none of the trials (out of 45) a linear Gaussian SSM is preferred over an SSM with GM process noise. This suggests that even when accounting for model complexity, the SSM with GM process noise fits the observed MEG data better than a linear Gaussian SSM, and can thus serve as a better explanatory model for the underlying biological processes.

## Discussion

We considered the problem of estimating latent dynamics of biological processes from noisy and limited observations, in which the commonly-used Gaussian statistics fail to capture the heterogeneous and switching nature of the dynamics. An instance of such dynamics are the neural processes that underlie auditory attention switching in a cocktail party consisting of multiple speakers. To address this shortcoming of Gaussian models, we utilized a SSM with GM process noise and devised an EM algorithm to estimate the parameters of the GM density from SSM observations. To approximate the intractable expectations in EM, we considered two approaches, one based on particle smoothing and another based on closed-form GM approximations to the smoothing densities.

The main limitation of the first approach based on particle smoothing is the exponential growth of the number of particles in terms of the dimension of the smoothing densities. The second approach based on closed-form GM approximations significantly reduces the computational complexity by requiring a cubic dependence in the state dimensions, with an additional cubic dependence in the number of GM mixtures. In addition, the closed-form approximations require a linear SSM model to hold. If the underlying state-space model is indeed non-linear, linearization techniques such as those used in the extended [1] or unscented Kalman filter [62] are required, which may result in model mismatch.

While both the observation and process noise are often non-Gaussian in practice, in our proposed framework, we have assumed Gaussian statistics to model the observation noise. This is motivated by the conventional preprocessing techniques applied to the observed data (e.g., source separation), which are able to remove the non-Gaussian noise components in such a way that the resulting 'denoised' observations admit Gaussian noise models. Nevertheless, the observation noise can also be modeled by a GM density, whose parameters can be estimated in a similar fashion to those of the process noise in our proposed framework. The resulting inference algorithm, however, would be more intricate and is deemed as a future extension of our current methodology.

As mentioned in the Introduction, existing parameter estimation techniques for general switching SSMs are computationally demanding and often require direct maximization of the data likelihood via numerical methods. The MSAR method presented here circumvents this challenge by using a surrogate of the states to perform parameter estimation. It is noteworthy that our SSM with GM process noise can be thought of as a special case of an SSM with underlying MSAR state dynamics, in which the state transition probability matrix is constrained to have equal rows. Thus, another potential extension of our proposed methodology is to utilize the closed-form approximation approach for estimating the parameters of more general switching SSMs in a computationally scalable fashion.

As our primary application, we considered the problem of dynamic TRF estimation from auditory neural responses to speech. We formulated the problem as a linear SSM with Gaussian or GM process noise, and compared the TRF estimates to those obtained by the RLS and MSAR algorithms. Application to simulated data shows that the algorithm can effectively recover the parameters of the underlying GM process noise and that the GM representation

improves state estimation for a synthesized latent process exhibiting heterogeneous and rapid dynamics. Application to experimentally-recorded MEG from both at-will and instructed attention switching two-speaker cocktail party settings revealed that the proposed SSM with GM process noise model and inference methodology clearly delineates the heterogeneous dynamics of the TRF components that are otherwise not captured by the other techniques. While the proposed methodology can be used as a reliable estimation technique for auditory attention decoding in a cocktail party settings, it can be applied to a wider range of biological problems in which the underlying model exhibits heterogeneous and switching dynamics.

## Methods

### Ethics statement

All experimental protocols and procedures were approved by the University of Maryland Institutional Review Board, and written informed consent was obtained from participants before the experiments.

### The main problem formulation

Consider the following generic discrete-time SSM with additive noise:

$$
\begin{cases}
\mathbf{x}_n = f_n(\mathbf{x}_{n-1}) + \mathbf{w}_n \\
\mathbf{y}_n = g_n(\mathbf{x}_n) + \mathbf{v}_n
\end{cases}
\tag{1}
$$

where $\mathbf{x}_n \in \mathbb{R}^{d_x}$ and $\mathbf{y}_n \in \mathbb{R}^{d_y}$ represent the states and the observations at time $n$, respectively. We assume that the functional forms of $f_n(.)$ and $g_n(.)$ are known and fixed for $n = 1, \ldots, N$, from domain-specific knowledge of the problem. Following our arguments in the Introduction on the utility of source separation techniques in removing the non-Gaussian components of the observation noise, we let $\mathbf{v}_n \sim \mathcal{N}(\mathbf{0}, \mathbf{R})$ be the i.i.d. Gaussian sequence of observation noise. The *process* noise $\mathbf{w}_n$, on the other hand, accounts for the stochasticity of the state evolution. Note that from a neuroscience perspective, the process noise consists of both the underlying behaviorally- and stimulus-driven dynamics as well as the background neural activity (not necessarily evoked by the stimulus or behavior). While the terminology alludes to a zero-mean Gaussian disturbance, the process noise in this context is typically quite structured and far from being a zero-mean Gaussian disturbance. Nevertheless, we adhere to this terminology for the sake of consistency with existing literature on state estimation.

To represent the process noise $\mathbf{w}_n$, consider a GM with $M$ mixture components and parameter set $\Theta \coloneqq \{p_{1:M}, \boldsymbol{\mu}_{1:M}, \Sigma_{1:M}\}$ containing the mixture probabilities $p_{1:M}$, mean vectors $\boldsymbol{\mu}_{1:M}$, and covariance matrices $\Sigma_{1:M}$. We model the state dynamics over $K \coloneqq N/W$ consecutive non-overlapping windows of length $W$. Within each window $i \in \{1, \ldots, K\}$, the process noise is drawn from one of the mixture components, which we denote by $z_i \in \{1, \ldots, M\}$. Therefore, we have $\mathbf{w}_n \sim \mathcal{N}(\boldsymbol{\mu}_{z_i}, \Sigma_{z_i})$ for $n = (i - 1)W + 1, \ldots, iW$, independent of $\mathbf{v}_n$, and we consider the $z_i$'s to be i.i.d. with $\mathrm{P}(z_i = m) = p_m$ for $m = 1, \ldots, M$. In other words, $z_i$ determines the active mixture component that governs the state dynamics in window $i$. This can also be interpreted as a switching Gaussian process noise. For the special case of $W = 1$, the resulting model could in principle approximate any arbitrary i.i.d. process noise $\mathbf{w}_n$, as it is fitting a GM model to the process noise. In this case, the labels $z_i$ of mixture components can vary at the same rate as that of the states and observations. Note that regardless of the choice of $W$, $\mathbf{w}_n$ has a GM distribution, but is only i.i.d. when $W = 1$. We hereafter refer to the foregoing process noise model, for general $W \geq 1$, as the *GM process noise model*.

Let $\mathcal{Y}_1^N$ denote the set of observations from 1 to $N$, i.e., $\mathbf{y}_{1:N}$, and similarly define $\mathcal{X}_1^N$ and $\mathcal{Z}_1^K$ for $\mathbf{x}_{1:N}$ and $z_{1:K}$, respectively. Our goal is to estimate the GM process noise parameters $\Theta$ from SSM observations $\mathcal{Y}_1^N$. As estimation of the observation noise covariance $\mathbf{R}$ in EM is straightforward [13], we assume $\mathbf{R}$ to be fixed for convenience and will briefly review the update equations for $\mathbf{R}$ in the following subsection, if it needs to be estimated from the observed data. As mentioned in the Introduction, $\mathbf{R}$ can also be estimated from stimulus-free conditions. Finally, we adopt the Maximum Likelihood (ML) estimation framework to estimate $\Theta$ as follows:

$$\hat{\Theta}_{\text{ML}} := \arg\max_{\Theta} P(\mathcal{Y}_1^N \mid \Theta). \tag{2}$$

Despite its simple statement, the problem of Eq (2) is challenging due to the difficulties in computing the optimization argument, i.e., data likelihood, in a computationally scalable fashion. We will address this challenge in the forthcoming section.

## Parameter estimation

We use the EM algorithm as a solution method for the ML problem in (2). The EM framework provides an iterative procedure to update the estimated parameter set with the guarantee that at iteration $(\ell + 1)$ we have

$$P(\mathcal{Y}_1^N \mid \hat{\Theta}^{(\ell+1)}) \geq P(\mathcal{Y}_1^N \mid \hat{\Theta}^{(\ell)}) \tag{3}$$

where $\hat{\Theta}^{(\ell)}$ is the parameter set estimate from the $\ell^{\text{th}}$ iteration [12]. The EM algorithm guarantees convergence to a local maximum, and most of the work on escaping the undesirable local maxima in EM theory have focused on providing an informed initialization of the algorithm [63, 64]. As explained in the Model Parameter Settings subsection of the Methods, we use the fixed-interval smoothed estimates based on a Gaussian model to choose $\hat{\Theta}^{(0)}$ and initialize the algorithm.

Let $\mathcal{H} = \{\mathcal{Z}_1^K, \mathcal{X}_1^N\}$ denote the set of latent variables in the SSM, which includes the states and the labels of active mixture component in each window. The EM algorithm performs the following two steps at the $(\ell + 1)^{\text{th}}$ iteration and repeats them until convergence to a parameter estimate $\hat{\Theta}$:

$$\begin{cases} \text{E-step}: Q(\Theta \mid \hat{\Theta}^{(\ell)}) = \mathrm{E}_{\mathcal{H}}\{\log P(\mathcal{Y}_1^N, \mathcal{H} \mid \Theta) \mid \mathcal{Y}_1^N, \hat{\Theta}^{(\ell)}\} \\ \text{M-step}: \hat{\Theta}^{(\ell+1)} = \arg\max_{\Theta} Q(\Theta \mid \hat{\Theta}^{(\ell)}) \end{cases} \tag{4}$$

where the surrogate function $Q(\Theta \mid \hat{\Theta}^{(\ell)})$ is a lower bound on the data log-likelihood. The expectation in the E-step is over the conditional density of $\mathcal{H} \mid \mathcal{Y}_1^N, \hat{\Theta}^{(\ell)}$. As all of the following expectations are also conditioned on $\mathcal{Y}_1^N$ and $\hat{\Theta}^{(\ell)}$, we drop the conditioning in the notation for convenience, but keep the expectation subscript to denote the random variable with respect to which the expectation is taken. Also, hereafter the subscript $(i, j)$ represents the time index of the $j^{\text{th}}$ sample in the $i^{\text{th}}$ window, i.e., $n = (i - 1)W + j$ for brevity. The EM algorithm in Eq (4) in our setting can be expressed as follows:

**E-step**: The surrogate function in the SSM is computed as

$$\begin{aligned} Q(\Theta \mid \hat{\Theta}^{(\ell)}) &= \mathrm{E}_{\mathcal{H}}\{\log P(\mathcal{Z}_1^K \mid \Theta) + \log P(\mathcal{X}_1^N \mid \mathcal{Z}_1^K, \Theta)\} + c_1 \\ &= \sum_{i=1}^{K}\sum_{m=1}^{M} \mathrm{E}_{\mathcal{H}}\left\{ \mathbb{1}_{\{z_i=m\}}\left(\log p_m + \sum_{j=1}^{W} \log \pi_{(i,j),m}\right)\right\} + c_2, \end{aligned} \tag{5}$$

where $\mathbb{1}_{\{.\}}$ denotes the indicator function, $c_1$ and $c_2$ are terms not dependent on $\Theta$, and $\pi_{(i, j), m}$ is defined as

$$\pi_{(i,j),m} := \mathrm{P}(\mathbf{x}_{(i,j)} \mid \mathbf{x}_{(i,j-1)}, z_i = m, \Theta), \tag{6}$$

which is computed based on the Gaussian density for $\mathbf{w}_{(i, j)}$ in Eq (1) when $z_i = m$. If we decompose the conditional expectation in Eq (5) into two iterated conditional expectations with respect to $\mathcal{X}_1^N \mid \mathcal{Y}_1^N, \hat{\Theta}^{(\ell)}$ and $\mathcal{Z}_1^K \mid \mathcal{X}_1^N, \hat{\Theta}^{(\ell)}$ (where $\mathcal{Y}_1^N$ is dropped in the latter due to conditional independence), this equation can be written as

$$Q(\Theta \mid \hat{\Theta}^{(\ell)}) = \sum_{i=1}^{K} \sum_{m=1}^{M} \mathrm{E}_{\mathcal{X}}\left\{\hat{\epsilon}_{i,m}^{(\ell)}\left(\log p_m + \sum_{j=1}^{W} \log \pi_{(i,j),m}\right)\right\} + c_3, \tag{7}$$

where $c_3$ is a constant and $\hat{\epsilon}_{i,m}^{(\ell)}$ is the membership probability and can be expressed using Bayes' rule as:

$$\hat{\epsilon}_{i,m}^{(\ell)} := \mathrm{P}\left(z_i = m \mid \mathcal{X}_1^N, \hat{\Theta}^{(\ell)}\right) = \frac{\hat{p}_m^{(\ell)} \prod_{j=1}^{W} \hat{\pi}_{(i,j),m}^{(\ell)}}{\sum_{m'=1}^{M} \hat{p}_{m'}^{(\ell)} \prod_{j=1}^{W} \hat{\pi}_{(i,j),m'}^{(\ell)}}, \tag{8}$$

The variable $\hat{\pi}_{(i,j),m}^{(\ell)}$ is defined similarly to (6) but for $\Theta = \hat{\Theta}^{(\ell)}$, which makes $\hat{\epsilon}_{i,m}^{(\ell)}$ a constant with respect to $\Theta$.

**M-step**: In this step, we maximize the log-likelihood lower bound with respect to $\Theta$. Differentiating (7) with respect to $\Theta$, enforcing the condition $\sum_{m=1}^{M} p_m = 1$, and invoking the dominated convergence theorem to change the order of expectation and differentiation, we obtain the following parameter updates for $m = 1, \ldots, M$:

$$\hat{p}_m^{(\ell+1)} = \frac{1}{K} \sum_{i=1}^{K} \mathrm{E}_{\mathcal{X}}\{\hat{\epsilon}_{i,m}^{(\ell)}\}, \quad \hat{\boldsymbol{\mu}}_m^{(\ell+1)} = \frac{\sum_{i=1}^{K} \mathrm{E}_{\mathcal{X}}\left\{\hat{\epsilon}_{i,m}^{(\ell)} \sum_{j=1}^{W} \mathbf{v}_{(i,j)}\right\}}{W \sum_{i=1}^{K} \mathrm{E}_{\mathcal{X}}\{\hat{\epsilon}_{i,m}^{(\ell)}\}}, \tag{9}$$

$$\hat{\boldsymbol{\Sigma}}_m^{(\ell+1)} = \frac{\sum_{i=1}^{K} \mathrm{E}_{\mathcal{X}}\left\{\hat{\epsilon}_{i,m}^{(\ell)} \sum_{j=1}^{W} \mathbf{v}_{(i,j)} \mathbf{v}_{(i,j)}^{\top}\right\}}{W \sum_{i=1}^{K} \mathrm{E}_{\mathcal{X}}\{\hat{\epsilon}_{i,m}^{(\ell)}\}} - \hat{\boldsymbol{\mu}}_m^{(\ell+1)} (\hat{\boldsymbol{\mu}}_m^{(\ell+1)})^{\top}, \tag{10}$$

where $\mathbf{v}_{(i, j)} := \mathbf{x}_{(i, j)} - f_{(i, j)}(\mathbf{x}_{(i, j-1)})$.

**Remark 1**. If the covariance matrix $\mathbf{R}$ of the Gaussian observation noise in (1) also needs to be estimated from $\mathcal{Y}_1^N$, it can be included in the parameter set $\Theta$. The update formula for $\hat{\mathbf{R}}^{(\ell+1)}$ in the EM framework then becomes [65]

$$\hat{\mathbf{R}}^{(\ell+1)} = \frac{1}{N} \sum_{n=1}^{N} \mathrm{E}_{\mathcal{X}}\{(\mathbf{y}_n - g_n(\mathbf{x}_n))(\mathbf{y}_n - g_n(\mathbf{x}_n))^{\top}\}. \tag{11}$$

In addition, if the function $f_n(\cdot)$ is only known in parametric form, it is in principle possible to estimate it via the same EM framework. As an example, which we use for TRF modeling, consider $f_n(\mathbf{x}) = \alpha\, \mathbf{x}$, where $\alpha$ is an unknown constant. Then, the coordinate descent update for

$\alpha$ takes the form [13]:

$$\hat{\alpha}^{(\ell+1)} = \frac{\sum_{i=1}^{K} E_{\mathcal{X}}\left\{ \hat{\epsilon}_{i,m}^{(\ell)} \sum_{j=1}^{W} \mathbf{x}_{(i,j-1)}^{\top} \hat{\Sigma}_{m}^{(\ell+1)-1} \mathbf{x}_{(i,j-1)} \right\}}{\sum_{i=1}^{K} E_{\mathcal{X}}\left\{ \hat{\epsilon}_{i,m}^{(\ell)} \sum_{j=1}^{W} \mathbf{x}_{(i,j-1)}^{\top} \hat{\Sigma}_{m}^{(\ell+1)-1} \left( \mathbf{x}_{(i,j)} - \hat{\boldsymbol{\mu}}_{m}^{(\ell+1)} \right) \right\}}. \tag{12}$$

In the definition of $\hat{\epsilon}_{i,m}^{(\ell)}$ in Eq (8), both the numerator and the denominator include exponential functions of the states. Therefore, the conditional expectations in Eq (7) and in the update equations above are intractable even if the joint smoothing density of $\mathcal{X}_1^N \mid \mathcal{Y}_1^N, \hat{\Theta}^{(\ell)}$ is known in closed-form [29]. In this work, we use two different approaches to address this challenge: In Approach 1, we use Monte Carlo Approximations for computing the aforementioned expectations. While this approach is rather straightforward to implement, the resulting algorithm is computationally intensive. In Approach 2, we instead derive closed-form approximations to the densities required for computing the expectations. The underlying parameters can be updated recursively, which makes the resulting algorithm scalable with the problem dimension. The details of these approaches are given in S1 Appendix.

## Dynamic estimation of the TRF

Consider a cocktail party setting [45], in which a subject is listening to two speakers simultaneously, but only attending to one of the speakers. While the subject is performing this task, the neural response is recorded using MEG. Let $y_t \in \mathbb{R}$ denote the auditory component of the neural response at time $t \in \{1, \ldots, T\}$, extracted from multichannel MEG recordings via the Denoising Source Separation (DSS) algorithm [22, 66]. Also, let $s_t^{(q)}$ be a speech feature of speaker $q \in \{1, 2\}$ at time $t$, e.g., the acoustic envelope, and denote by $\mathbf{s}_t^{(q)} = [s_t^{(q)}, \ldots, s_{t-L-1}^{(q)}]^{\top} \in \mathbb{R}^L$ the vector containing the previous $L$ features up to (and including) time $t$. In this work, we consider $s_t^{(q)}$ to be the acoustic envelope in log scale, which is known to be a reliable predictor of the neural response [47]. Other features such as phoneme representations, word frequency measures, and semantic composition have also been considered in the literature [52–54], and can also be included in $s_t^{(q)}$. A widely-used linear stimulus-response model is given by:

$$y_t = \mathbf{s}_t^{\top} \tilde{\boldsymbol{\tau}}_t + v_t, \tag{13}$$

where $\tilde{\boldsymbol{\tau}}_t = [\tilde{\boldsymbol{\tau}}_t^{(1)}; \tilde{\boldsymbol{\tau}}_t^{(2)}] \in \mathbb{R}^{2L}$ is the concatenation of $\tilde{\boldsymbol{\tau}}_t^{(1)}$ and $\tilde{\boldsymbol{\tau}}_t^{(2)}$ as the TRFs at time $t$ corresponding to speakers 1 and 2, respectively. Also, $\mathbf{s}_t = [\mathbf{s}_t^{(1)}; \mathbf{s}_t^{(2)}] \in \mathbb{R}^{2L}$ is the concatenation of the speech feature vectors at time $t$, and $v_t$ represents the observation noise. In light of this model, and as mentioned in the Introduction, the TRF $\tilde{\boldsymbol{\tau}}_t^{(q)}$ can be thought of as the impulse response of a linear, but time-varying, system representing the neural activity and taking as input the speech features of speaker $q$, for $q = 1, 2$.

We assume $v_t \sim \mathcal{N}(0, \sigma^2)$ and define the nominal observation SNR as $10 \log_{10}(\bar{E}/\sigma^2)$, where $\bar{E}$ is the average of the signal component in Eq (13) over the trial of length $T$. It is common to consider a piecewise-constant approximation to the TRFs over consecutive non-overlapping time windows of length $t_0$, which is comparable to the length of the TRF $L$. In other words, $\tilde{\boldsymbol{\tau}}_t = \boldsymbol{\tau}_n$ for $t \in \{(n-1)t_0 + 1, \ldots, nt_0\}$ and $n \in \{1, \ldots, N\}$ where $N = T/t_0$ is assumed to be an integer without loss of generality. We then define $\mathbf{y}_n = [y_{(n-1)t_0+1}, \ldots, y_{nt_0}]^{\top}$, $\mathbf{S}_n = [\mathbf{s}_{(n-1)t_0+1}, \ldots, \mathbf{s}_{nt_0}]$, and $\mathbf{v}_n = [v_{(n-1)t_0+1}, \ldots, v_{nt_0}]^{\top}$.

**TRF estimation via regularized RLS.** First, the TRFs are represented over a dictionary $\mathbf{G}$, i.e, $\boldsymbol{\tau}_n^{(q)} = \mathbf{G}\mathbf{x}_n^{(q)}$, in order to enforce smoothness in the lag domain and to mimic static TRF estimates [48, 49]. The dynamic TRF estimation framework of [47] can be stated as:

$$
\begin{cases}
\hat{\mathbf{x}}_n = \arg\ \min_{\mathbf{x}\in\mathbb{R}^{2L}} \sum_{i=1}^{n} \lambda^{n-i} \parallel \mathbf{y}_i - \mathbf{S}_i^\top \tilde{\mathbf{G}}\mathbf{x} \parallel_2^2 + \gamma h(\mathbf{x}) \\[2mm]
\hat{\boldsymbol{\tau}}_n = \tilde{\mathbf{G}}\hat{\mathbf{x}}_n
\end{cases}
\tag{14}
$$

where $\lambda \in (0, 1)$ is the forgetting factor, $\gamma$ is the regularization coefficient, $h(.)$ can either be an $\ell_1$ or $\ell_2$ penalty [57], and $\tilde{\mathbf{G}} = \operatorname{diag}(\mathbf{G}, \mathbf{G})$ is a block diagonal matrix with $\mathbf{G}$ containing the dictionary atoms. Similar to [11, 47], we consider a Gaussian dictionary $\mathbf{G} \in \mathbb{R}^{L\times D}$ where the $D$ columns of $\mathbf{G}$ are shifted Gaussian kernels. The parameter $\lambda$ in Eq (14) induces a trade-off between adaptivity and robustness of TRF estimation.

**TRF estimation via MSAR modeling.** While the RLS estimates of the TRF capture the dynamics via the forgetting factor mechanism, they are not capable of capturing abrupt and/or recurring state dynamics. MSAR models, on the other hand, explicitly model such dynamics and are thus a suitable class of models for TRF estimation. Given that the TRF is not directly observable, the conventional MSAR models are not readily applicable. In addition, the SSM extensions of MSAR models do not admit simple parameter estimation procedures. We thus consider the regularized least squares (LS) estimates of the TRFs, i.e., the RLS estimates with $\lambda = 0$, as a surrogate of the true TRFs, which can then be modeled as an MSAR process.

To this end, let $\hat{\mathbf{x}}_n$ be the regularized LS estimates of the TRF. To capture the dynamics of $\hat{\mathbf{x}}_n$, we consider a first-order Markov-switching process with $J$ states. The underlying HMM is parameterized by the initial probabilities $\pi_i$, $i = 1, 2, \cdots, J$ and transition probability matrix $P_{ij}$, $i, j = 1, 2, \cdots, J$. Let $s_n \in \{1, 2, \cdots, J\}$ denote the state at time $n$. Then, we have:

$$
\hat{\mathbf{x}}_n = \alpha_j \hat{\mathbf{x}}_{n-1} + \mathbf{w}_{j,n}, \quad \text{if } s_n = j, \ j = 1, 2, \cdots, J,
\tag{15}
$$

where $\alpha_j$ is the rate of change of the TRF in state $j$, and $\mathbf{w}_{j,n} \sim \mathcal{N}(\boldsymbol{\mu}_j, \mathbf{Q}_j)$ is the i.i.d. sequence of process noise in state $j$, $j = 1, 2, \cdots, J$. The parameters to be estimated are $\mathcal{M} := \{\{\pi_i\}_{i=1}^{J}, \{P_{ij}\}_{i,j=1}^{J,J}, \{\alpha_j, \boldsymbol{\mu}_j, \mathbf{Q}_j\}_{j=1}^{J}\}$. Let $\omega_{j,n}$ denote $P[s_n = j|\{\hat{\mathbf{x}}_m\}_{m=1}^{n}, \mathcal{M}]$. Then, the MSAR estimates are given by:

$$
\hat{\mathbf{x}}_n^{(\text{MSAR})} := \sum_{j=1}^{J} \omega_{j,n}(\boldsymbol{\mu}_j + \alpha_j \hat{\mathbf{x}}_{n-1}), \quad n = 1, 2, \cdots, N.
\tag{16}
$$

In S1 Appendix, we provide an EM-based algorithm for estimating the parameters $\mathcal{M}$ and recursively computing $\omega_{j,n}$.

**TRF estimation via state-space models.** The RLS estimate in (14) is a filtering estimate by design and is suited for real-time estimation of TRFs. For a more precise dynamic analysis of the TRFs in an off-line fashion, SSMs have the advantage of providing smoothed estimates and directly modeling the evolution of the TRFs through the state equation. We use the SSM below to represent the TRF dynamics and its relation to the neural response:

$$
\begin{cases}
\mathbf{x}_n = \alpha\mathbf{x}_{n-1} + \mathbf{w}_n \\[1mm]
\boldsymbol{\tau}_n = \tilde{\mathbf{G}}\mathbf{x}_n \\[1mm]
\mathbf{y}_n = \mathbf{S}_n^\top \boldsymbol{\tau}_n + \mathbf{v}_n
\end{cases}
\tag{17}
$$

where $\alpha \in (0, 1)$ controls the nominal rate of change of the TRF, similar to the effect of the forgetting factor $\lambda$ in Eq (14) for the RLS framework. In [67], a correspondence between $\alpha$ and $\lambda$ has been discussed which can result in the same filtering estimates of the SSM in Eq (17) with Gaussian noise and the RLS model in Eq (14), without any penalization. The parameter $\alpha$ can either be estimated in the EM framework as in Eq (12) [13], or it can be set based on the domain-specific knowledge of the problem to provide a target adaptivity-robustness trade-off, akin to choosing the forgetting factor in the RLS algorithm. The estimated TRFs in (17) are computed from the smoothing estimates as $\hat{\boldsymbol{\tau}}_n = \tilde{\mathbf{G}}\hat{\mathbf{x}}_{n|N}$.

By assuming a GM density for $\mathbf{w}_n$ in Eq (17), we can similarly obtain smoothing estimates of the TRFs, by using the two approaches discussed in the preceding section.

## Model parameter settings

The following subsections provide detailed information on the choice of the various model parameters used in the simulation study and application to experimentally-recorded data from the Results section.

**Parameter settings of the simulation study.** For the simulation study, we use a sampling rate of $F_s = 100$ Hz and a length of 250 ms for the TRFs, i.e., $L = 0.25F_s$. Let $\mathbf{G}$ be a dictionary consisting of five Gaussian atoms with variances of 0.018 s$^2$ whose means are separated by 50 ms increments starting from a lag of 0 ms to 200 ms. This results in $\mathbf{G} \in \mathbb{R}^{25 \times 5}$ and $\mathbf{x}_n \in \mathbb{R}^{10}$ in Eqs (14) and (17). We consider a piecewise-constant model for the TRFs over windows of length 300 ms resulting in $N = 300$ TRF samples over the trial for each speaker.

We consider $W = 5$, i.e., the TRF dynamics are governed by one mixture component in each window of length $Wt_0/F_s = 1.5$ s. For simplicity, we consider $\Sigma_{1:M}$ to be diagonal, which makes the parameter update formulas of Eqs. (S3) and (S5) in S1 Appendix to also take diagonal forms. The number of mixture components is chosen as $M = 5$ using the AIC criterion and log-likelihoods computed using Eqs. (S19) and (S20) given in S1 Appendix. The number of states $J$ in the MSAR model can also be chosen via AIC, but we here take $J$ to be the same as $M$ for fairness of comparison with the SSM model with GM process noise.

We also set the parameters of Algorithm S2 given in S1 Appendix as $\Gamma_F = \Gamma_B = \Gamma_S = M$. To initialize the EM algorithm, we use two methods: 1) initializing with $\hat{p}_{1:M}^{(0)} = \frac{1}{M}$, random means $\hat{\boldsymbol{\mu}}_{1:M}^{(0)}$ close to zero, and $\hat{\Sigma}_{1:M}^{(0)}$ equal to the estimated process noise covariance in the linear Gaussian SSM, and 2) setting $\hat{\Theta}^{(0)}$ as the GM fit to the empirical samples of process noise in the linear Gaussian SSM, which are computed from the smoothed state estimates. In other words, a GM is fit on the state residuals, i.e., empirical process noise samples, $\hat{\mathbf{w}}_n = \hat{\mathbf{x}}_n^{(s)} - \alpha\hat{\mathbf{x}}_{n-1}^{(s)}$ where $\hat{\mathbf{x}}_n^{(s)}$ denotes the smoothed states using a linear Gaussian SSM. The state residuals here do not necessarily exhibit a clear multimodal histogram due to the Gaussian assumption in the model and the inaccuracies in state estimation. Nevertheless, a GM fit on the state residuals serves as a reasonable initialization for the EM algorithm in our experience.

Note that in the simulation studies, we have used the first initialization strategy to show that under reasonable SNR conditions, the algorithm is able to initialize with large covariances, i.e., based on the linear Gaussian SSM estimates, and subsequently retrieve the concentrated mixture components. This is analogous to particle smoothing methods where the initial samples are drawn from a broad density and through consecutive weighting and resampling, the particles can eventually capture the underlying densities. In our experience, the second initialization strategy results in faster convergence, especially under poor SNR conditions, due to the extra information extracted from the residual estimates from the linear Gaussian SSM. Thus, for the real data analysis, we have used the second initialization strategy.

For the forgetting factor λ in RLS, an effective estimation length [47] of 2 s is chosen to result in comparable TRF estimates to those of the SSM with $\alpha = 0.99$. Also, $\gamma$ in Eq (14) for an $\ell_2$ penalty is tuned through two-fold cross-validation. For the Gaussian SSM and the SSM with GM process noise, diagonal process noise covariance matrices are considered, and both the process and observation noise parameters as well as the states are estimated simultaneously for each trial run.

We have considered a total of $U = 2000$ particles in Algorithm S1 given in S1 Appendix to approximate densities of dimension $2D(W + 1) = 60$, so that state estimates are comparable to those obtained by the closed-form approximation. Note that the choice of the number of particles is critical for the performance of particle smoothing, as the number of particles required for stable estimation grows exponentially in the dimension of the densities.

**Parameter settings of the experimentally-recorded data analysis.**   We set the TRF length to 300 ms and consider TRFs to be piece-wise constant over windows of length 400 ms. Also, we choose $W = 5$ to enforce that the TRF dynamics are governed by one mixture component in each window of length 2 s. We represent the TRFs over a Gaussian dictionary with means separated by 20 ms starting from 0 to 280 ms, and variances of $8.5 \times 10^{-3}$ s$^2$. The parameters λ and $\alpha$ are set to 0.92 and 0.97, respectively, to achieve comparable TRF estimates from Eqs (14) and (17). The $\ell_2$ penalty $\gamma$ in (14) is determined via two-fold cross-validation. We consider diagonal covariance matrices for the process noise to reduce the dimension of Θ, and estimate the observation noise $\sigma^2$ in the EM framework. The forgetting factor in Eq (14) enforces a temporal continuity in TRF estimates and increases robustness to noise and artifacts. The same effect can be replicated in the SSM of Eq (17) by considering $\alpha$ close to one and restricting the dynamic range of the process noise $\mathbf{w}_n$.

To enforce the latter, we consider IG conjugate priors [61] on the diagonal elements of the process noise covariance matrices. For the Gaussian SSM with $\mathbf{w}_n \sim \mathcal{N}(\mathbf{0}, \mathbf{Q})$ and $\mathbf{Q} = \text{diag}([q_1, \ldots, q_{2D}])$, the log-prior takes the form

$$\kappa \log \mathrm{P}(\mathbf{Q}) = \kappa \sum_{d=1}^{2D} ((\tilde{\alpha}_d + 1) \log q_d + \tilde{\beta}_d / q_d) + c_4, \qquad (18)$$

where $\{\tilde{\alpha}_d, \tilde{\beta}_d\}_{d=1}^{2D}$ are the parameters of the IG prior and $c_4$ includes terms not dependent on $q_d$'s. The log-prior is then added to the surrogate Q-function of the EM algorithm, and $\kappa$ determines the strength of the prior with respect to the complete data log-likelihood. We choose $\kappa = N$ for the linear Gaussian case and $\kappa = N/M$ for the linear SSM with GM process noise, to correct for the number of mixture components. We tune the IG parameters using empirical samples of the process noise from the RLS estimates, computed as $\hat{\mathbf{w}}_n = \hat{\mathbf{x}}_n^{(\mathrm{RLS})} - \alpha \hat{\mathbf{x}}_{n-1}^{(\mathrm{RLS})}$. Thus, the process noise variance is controlled by the IG prior, which prohibits drastic temporal changes in the TRF. For the SSM with GM process noise, we also bound the elements of $\hat{\boldsymbol{\mu}}_{1:M}^{(\ell)}$ in each EM iteration such that the variance of the estimated GM process noise along each dimension is not larger than those of the linear Gaussian case, i.e., estimated $q_d$'s using the EM algorithm. Note that in the absence of such priors, the EM algorithm would likely result in TRFs that are highly variable in time and with no meaningful morphological structure.

## Subjects, stimuli, and procedures

We have used data from two separate attention switching experiments in this work, which we refer to as the at-will and instructed attention switching experiments. Neuromagnetic signals were recorded at a sampling frequency of 2 kHz using a 157-sensor whole-head MEG system (Kanazawa Institute of Technology, Nonoichi Ishikawa, Japan) in a dim magnetically shielded room.

The at-will attention switching dataset is a subset of recordings in [51], where the participants included five younger-adult (22-33 years old) native English speakers with normal hearing recruited from the University of Maryland. Only one of the subjects exhibited a meaningful auditory neural response (i.e., auditory DSS rotation matrix; see MEG Data Preprocessing subsection for details) with a reliable behavioral report. Two stories were presented diotically to subjects' ears, one narrated by a male speaker and the other one by a female speaker. The stimuli consisted of two segments from the book, *The Legend of Sleepy Hollow* by Washington Irving. Subjects listened to trials of the same speech mixture (each 90 s in duration), and were instructed to start attending to the male speaker first, and then to switch their attention between the two speakers at their own will for a minimum of one and a maximum of three times during each trial. Subjects were also given a switching button that they were instructed to press every time they decided to switch attention. For each subject, 3 trials were recorded. Prior to the experiment, a single-speaker pilot study was performed where subjects listened to three 60 s trials with similar stimuli. Further experimental details can be found in [51].

The instructed attention switching dataset is from the recordings in [50], where participants included seven normal hearing young adults (20-31 years old). The stimuli consist of four segments from the book *A Child's History of England* by Charles Dickens narrated by a male and female reader. Two different 60 s-long speech mixtures of the two speakers were generated, and each mixture was presented to subjects diotically for three trials. In each trial, subjects were instructed to focus on one speaker in the first 28 s of the trial, switch their attention to the other speaker after hearing a 2 second pause (between 28 s and 30 s time stamps), and maintain their focus on the latter speaker through the end of the trial. After completing the trials for each mixture, subjects answered comprehensive questions related to the passages they attended to. The MEG recording and preprocessing setup for this experiment is similar to that of the at-will attention switching experiment, and more details can be found in [50].

**MEG data preprocessing.**   Three reference channels were used to measure and cancel the environmental magnetic field by using time-shift PCA [21]. All MEG channels and speech envelopes were band-pass filtered between 2 Hz and 8 Hz (delta and theta bands), corresponding to the slow temporal modulations in speech [46, 48], and downsampled to $F_s$ = 100 Hz. Similar to [47, 50, 51], we used the DSS algorithm [22] on pilot trials to decompose the MEG data into temporally uncorrelated components. By using an averaging bias filter for promoting consistency across trials, we ordered the DSS components according to their trial-to-trial phase-locking reliability and chose the first component as the auditory neural response.

## Supporting information

**S1 Appendix. Supplementary methods.** This appendix includes: (i) detailed derivations of the two approaches used for computing the expectations in Eqs (9), (10), and (11), (ii) the criteria for model order selection, and (iii) details of the MSAR estimation procedure. (PDF)

## Author Contributions

**Conceptualization:** Jonathan Z. Simon, Michael C. Fu, Steven I. Marcus, Behtash Babadi.

**Data curation:** Alessandro Presacco.

**Formal analysis:** Sina Miran, Behtash Babadi.

**Funding acquisition:** Jonathan Z. Simon, Michael C. Fu, Steven I. Marcus, Behtash Babadi.

**Investigation:** Sina Miran, Jonathan Z. Simon, Steven I. Marcus, Behtash Babadi.

**Methodology:** Sina Miran, Alessandro Presacco, Jonathan Z. Simon, Michael C. Fu, Steven I. Marcus, Behtash Babadi.

**Project administration:** Behtash Babadi.

**Resources:** Jonathan Z. Simon, Behtash Babadi.

**Software:** Sina Miran.

**Supervision:** Jonathan Z. Simon, Steven I. Marcus, Behtash Babadi.

**Validation:** Sina Miran, Alessandro Presacco, Jonathan Z. Simon, Behtash Babadi.

**Visualization:** Sina Miran.

**Writing – original draft:** Sina Miran, Alessandro Presacco, Jonathan Z. Simon, Michael C. Fu, Steven I. Marcus, Behtash Babadi.

**Writing – review & editing:** Sina Miran, Alessandro Presacco, Jonathan Z. Simon, Michael C. Fu, Steven I. Marcus, Behtash Babadi.

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
