## [Decision Letter · Decision Letter 0]

22 Apr 2020

Dear Prof. Babadi,

Thank you very much for submitting your manuscript "Dynamic Estimation of Auditory Temporal Response Functions via State-Space Models with Gaussian Mixture Process Noise" for consideration at PLOS Computational Biology.

As with all papers reviewed by the journal, your manuscript was reviewed by members of the editorial board and by several independent reviewers. In light of the reviews (below this email), we would like to invite the resubmission of a significantly-revised version that takes into account the reviewers' comments.

The technical quality has generally been very appreciated, even though some issues need to be addressed.

An aspect necessary to work on is the (neuro)biological relevance.

We cannot make any decision about publication until we have seen the revised manuscript and your response to the reviewers' comments. Your revised manuscript is also likely to be sent to reviewers for further evaluation.

Sincerely,

Daniele Marinazzo

Deputy Editor

PLOS Computational Biology

Daniele Marinazzo

Deputy Editor

PLOS Computational Biology

Reviewer's Responses to Questions

**Comments to the Authors:**

Reviewer #1: Summary:

The paper introduces a Gaussian Mixture (GM) state space method (SSM) designed to estimate the latent dynamics of neural systems.

The method is particularly suitable in experimental settings were sudden switches between neural states are expected since they induces multi-modal distributions that are well-captured by GM models.

one of these settings is dichotic listening, where the participant can switch their attention towards one of two simultaneously presented speech recordings.

The core of the paper is a family of expectation-maximization EM algorithms for efficient maximum likelihood inference. The proposed algorithms cleverly are well designed and properly validated. The proposed method is definitely relevant for cognitive neuroscience as multi-modality is common in the analysis of brain signals and it can lead to meaningful interpretation of the data in terms of simpler latent signals.

In general, while I highly appreciate the technical quality of the paper, I think that at the present state it is more suitable for a machine learning/sigma processing venue. However, the paper could fit in PLOS comp bio if the authors put some effort in improving the exposition and stressing the neuroscientific motivations.

Major Comments:

1) I think that the structure of the paper is not very suitable to the PLOS comp bio audience. The core of the paper is purely methodological without much reference to neuroscience. The method section shoud ideally be placed after the results and some of the most technical details should be moved to supplementary material. The result section should begin a high level treatment of the method focused on its underlying assumptions and its connections with alternative methods commonly used in neuroscience (i.e. TRF). This will allow the less technically minded reader to understand the main results without getting bogged on technical details that are of higher interest for the machine learning and statistics audience. Emphasis should be given on the formulation and justification of the model instead of the algorithmic solution.

2) The results of the method should be compared with a wider range of baseline models. In particular, there need to be a quantitative comparison with Markov-switching autoregressive models as they are a obvious alternative to the GM approach. These models should be also discussion in the introduction.

Minor comments:

1) Referring to w_n as noise in Eq.1 is rather confusing from a neuroscience point of view as it actually represent the cognitively modulated signal. This terminology is particularly confusing since w_n has a non vanishing expected value.

Reviewer #2: “Dynamic Estimation of Auditory Temporal Response Functions via State-Space Models with Gaussian Mixture Process Noise” by Miran and colleagues tackles the estimation of the neural dynamics underlying auditory process from MEG recordings in a cocktail party setting. The authors develop a method based on Expectation-Maximization algorithm to infer the parameters of a Gaussian mixture process noise from state-space observations. The method is applied both to simulated and real MEG data. They state that representing the process noise as a Gaussian mixture process significantly improves state estimation and the underlying dynamics. Finally, application to MEG data reveals improvements over existing techniques that estimate the Temporal Response Functions (TFR).

However the study is well introduced respect to previous literature, some reference about application of state space model to MEG data misses (see e.g. Sorrentino et al, 2009, 2014, Sommariva 2014).

Overall the article is well written and provides a lot of mathematical details about the developed algorithms, some of which could be moved to Appendix to facilitate the reading. Indeed sometime the reader could loose the final aim to estimate the parameters of the Gaussian Mixture process noise.

I have very few comments:

- line 112: "To represent the process noise" shoud be To represent the process noise wn

- In line 126 for the sake of clarity should be better replacing n1, n2 with 1 to N

- some information on computational time on a generic machine could be added

- in both simulated and read data, a discussion on sensitivity to chosen parameters (e.g. M, number of particles) could be given

- an overall analysis of the drawbacks of the method could be added

Reviewer #3: The authors present a state-space model in which the process noise is modeled by a mixture of Gaussians. They derive the EM updating equations, where some of the variables turn out to be intractable. The authors then provide two algorithms to estimate these: one based on Monte-Carlo and one based on approximative closed-form expressions based on Taylor approximations. The method is applied in a dynamic TRF estimation problem. In simulations, both approaches are almost equally accurate, whereas the latter is much faster in terms of computations.

The paper is interesting and very well written. I only have one major comment (and several minor ones, see below).

While the simulations are convincing, I find the results provided on real data a bit too anecdotal (two trials of a single subject). It would be fine as an illustrative example of a potential application for the proposed algorithm, but as the title does emphasize TRF estimation as the focus of the paper, I feel the real-data experiments are not that convincing. Would it be possible to do a group analysis and to show that the GM-based model results in a statistically significant improvement wrt RLS or a linear Gaussian SSM? I know it is hard due to lack of a ground truth TRF, but perhaps the attention decoding accuracy could be used as an indirect performance metric. A group-wise version of Fig. 6E may also be interesting (after aligning the attention switches of the different subjects), showing medians and quantiles across subjects.

Detail comments:

- In the introduction the authors motivate the use of GMs for the process noise, while claiming that a single Gaussian is sufficient to model measurement noise. I don't fully agree with that statement. In the case of neural signals there are often artifacts, which may result in outliers in the signals which typically do not follow Gaussian statistics. Moreover, the noise process in (30) is background MEG activity (unrelated to the stimulus response). Background MEG is known to change over time, i.e., different covariance matrices R would be 'active' in different segments of the data. So it would also make sense to model measurement noise in a similar fashion as proposed for the process noise. Can the method be extended to also include gaussian mixtures in the observation noise? (I am not asking to include this, but rather to briefly comment on this case and how difficult/trivial it would be to add this extension).

-The SMCEM acronym is not explained in intro

-Which bias filter was used in the DSS method? Usually repeated stimuli are required, but the description of the experiment protocol does not mention any repeated stimuli. Please elaborate.

-page 19: I guess y_t is one of the DSS components? Please explain this.

-it is mentioned that the parameter alpha can be estimated within the EM framework. It may be useful for the reader to have the corresponding updating equation (could be added in an appendix). In fact, it could be useful to provide all the updating equations for the model defined in (32) as a special case of the more general updating equations.

- it is mentioned that -alternatively- the alpha parameter can be set based on the domain-specific knowledge of the problem to provide a desired adaptivity-robustness trade-off. I find this statement a bit strange, as the state dynamics are defined by the dynamics of the underlying (neural) processes. The word 'desired' therefore seems a bit strange here. Although I do understand what the authors mean (this parameter is indeed akin to the forgetting factor in RLS), I would rephrase or elaborate a bit.

-the second strategy to initialize the EM process (page 24, line375): Is this based on the oracle gaussian fit in Fig. 2-C? Or do you mean to first run a naive model with a single Gaussian? Please explain this a bit better.

- It is mentioned that the results of Fig. 3 are based on the first initialization strategy for the EM procedure. Is that also the case for Fig. 4 and 5? Are there no results for the second initialization strategy?

-Fig. 5: it would help if the ground truth TRF heatmap is also added here for convenience

-In the last sentence of the 'results' section, the authors provide attention decoding accuracies. How are these defined? Are they computed over the entire trial on a sample-by-sample basis?

- The description of the data set mentions five subjects, yet only one subject is reported in the results.

-Typo: Although the estimated process noise variance in in the GM case is controlled by...  2x 'in'

**Have all data underlying the figures and results presented in the manuscript been provided?**

Reviewer #1: Yes

Reviewer #2: Yes

Reviewer #3: Yes

PLOS authors have the option to publish the peer review history of their article (what does this mean?). If published, this will include your full peer review and any attached files.

Reviewer #1: Yes: Luca Ambrogioni

Reviewer #2: No

Reviewer #3: Yes: Alexander Bertrand
---

## [Decision Letter · Decision Letter 1]

21 Jul 2020

Dear Prof. Babadi,

We are pleased to inform you that your manuscript 'Dynamic Estimation of Auditory Temporal Response Functions via State-Space Models with Gaussian Mixture Process Noise' has been provisionally accepted for publication in PLOS Computational Biology.

Best regards,

Daniele Marinazzo

Deputy Editor

PLOS Computational Biology

Daniele Marinazzo

Deputy Editor

PLOS Computational Biology

Reviewer's Responses to Questions

**Comments to the Authors:**

Reviewer #2: My previous comments have been sufficiently addressed. The manuscript was significantly improved and in my opinion could be accepted in the present form.

Best,

Annalisa Pascarella

Reviewer #3: The authors have adequately addressed my comments and modified the paper accordingly. It is ready for publication. Congratulations on this nice work.

One final suggestion (just as a side remark, I am not requesting any changes in the manuscript here). Concerning the issue with poor DSS outcomes on 4/5 subjects: it may be worth to also try the stimulus-aware spatial filtering method from Das et al., which is akin to DSS, but it can achieve higher SNRs and it does not use repeated trials for the creation of a bias filter.

Das et al. "Stimulus-aware spatial filtering for single-trial neural response and temporal response function estimation in high-density EEG with applications in auditory research", NeuroImage, Vol. 204, 116211, 2020.

**Have all data underlying the figures and results presented in the manuscript been provided?**

Reviewer #2: None

Reviewer #3: Yes

PLOS authors have the option to publish the peer review history of their article (what does this mean?). If published, this will include your full peer review and any attached files.

Reviewer #2: **Yes: **Annalisa Pascarella

Reviewer #3: **Yes: **Alexander Bertrand

---

## [Editor Report · Acceptance letter]

12 Aug 2020

PCOMPBIOL-D-20-00251R1 

Dynamic Estimation of Auditory Temporal Response Functions via State-Space Models with Gaussian Mixture Process Noise

Dear Dr Babadi,

I am pleased to inform you that your manuscript has been formally accepted for publication in PLOS Computational Biology. Your manuscript is now with our production department and you will be notified of the publication date in due course.

With kind regards,

Matt Lyles
